# A Function Space View of Bounded Norm Infinite Width ReLU Nets: The Multivariate Case

**Greg Ongie**
Department of Statistics
University of Chicago
Chicago, IL 60637, USA
`gongie@uchicago.edu`

**Rebecca Willett**
Department of Statistics & Computer Science
University of Chicago
Chicago, IL 60637, USA
`willett@uchicago.edu`

**Daniel Soudry**
Electrical Engineering Department
Technion, Israel Institute of Technology
Haifa, Israel
`daniel.soudry@technion.ac.il`

**Nathan Srebro**
Toyota Technological Institute at Chicago
Chicago, IL 60637, USA
`nati@ttic.edu`

## Abstract

We give a tight characterization of the (vectorized Euclidean) norm of weights required to realize a function $f : \mathbb{R}^d \to \mathbb{R}$ as a single hidden-layer ReLU network with an unbounded number of units (infinite width), extending the univariate characterization of Savarese et al. (2019) to the multivariate case.

## 1 Introduction

It has been argued for a while, and is becoming increasingly apparent in recent years, that in terms of complexity control and generalization in neural network training, "the size [magnitude] of the weights is more important then the size [number of weights or parameters] of the network" (Bartlett, 1997; Neyshabur et al., 2014; Zhang et al., 2016). That is, inductive bias and generalization are not achieved by limiting the size of the network, but rather by explicitly (Wei et al., 2019) or implicitly (Nacson et al., 2019; Lyu & Li, 2019) controlling the magnitude of the weights.

In fact, since networks used in practice are often so large that they can fit any function (any labels) over the training data, it is reasonable to think of the network as virtually infinite-sized, and thus able to represent essentially all functions. Training and generalization ability then rests on fitting the training data while controlling, either explicitly or implicitly, the magnitude of the weights. That is, training searches over all functions, but seeks functions with small *representational cost*, given by the minimal weight norm required to represent the function. This "representational cost of a function" is the actual inductive bias of learning—the quantity that defines our true model class, and the functional we are actually minimizing in order to learn. Understanding learning with overparameterized (virtually infinite) networks thus rests on understanding this "representational cost", which is the subject of our paper. Representational cost appears to play an important role in generalization performance; indeed Mei & Montanari (2019) show that minimum norm solutions are optimal for generalization in certain simple cases, and recent work on "double descent" curves is an example of this phenomenon (Belkin et al., 2019; Hastie et al., 2019).

We can also think of understanding the representational cost as asking an approximation theory question: what functions can we represent, or approximate, with our de facto model class, namely the class of functions representable with small magnitude weights? There has been much celebrated work studying approximation in terms of the network *size*, *i.e.*, asking how many units are necessary in order to approximate a target function (Hornik et al., 1989; Cybenko, 1989; Barron, 1993; Pinkus, 1999). But if complexity is actually controlled by the norm of the weights, and thus our true model class is defined by the magnitude of the weights, we should instead ask how large a norm is necessary in order to capture a target function. This revised view of approximation theory should also change how we view issues such as depth separation: rather then asking how increasing depth can reduce

the *number* of units required to fit a function, we should instead ask how increasing depth can reduce the *norm* required, *i.e.*, how the representational cost we study changes with depth.

Our discussion above directly follows that of Savarese et al. (2019), who initiated the study of the representational cost in term of weight magnitude. Savarese et al. considered two-layer (*i.e.*, single hidden layer) ReLU networks, with an unbounded (essentially infinite) number of units, and where the overall Euclidean norm (sum of squares of all the weights) is controlled. (Infinite width networks of this sort have been studied from various perspectives by *e.g.*, Bengio et al. (2006); Neyshabur et al. (2015); Bach (2017); Mei et al. (2018)). For univariate functions $f : \mathbb{R} \to \mathbb{R}$, corresponding to networks with a single one-dimensional input and a single output, Savarese et al. obtained a crisp and precise characterization of the representational cost, showing that minimizing overall Euclidean norm of the weights is equivalent to fitting a function by controlling:

$$\max \left( \int |f''(x)| dx, |f'(-\infty) + f'(+\infty)| \right). \tag{1}$$

While this is an important first step, we are of course interested also in more than a single one-dimensional input. In this paper we derive the representational cost for any function $f : \mathbb{R}^d \to \mathbb{R}$ in any dimension $d$. Roughly speaking, the cost is captured by:

$$\|f\|_{\mathcal{R}} \dot{\approx} \|\mathcal{R}\{\Delta^{(d+1)/2} f\}\|_1 \approx \|\partial_b^{d+1} \mathcal{R}\{f\}\|_1 \tag{2}$$

where $\mathcal{R}$ is the Radon transform, $\Delta$ is the Laplacian, and $\partial_b$ is a partial derivative w.r.t. the offset in the Radon transform (see Section 3 for an explanation of the Radon transform). This characterization is rigorous for odd dimensions $d$ and for functions where the above expressions are classically well-defined (*i.e.*, smooth enough such that all derivatives are finite, and the integrand in the Radon transform is integrable). But for many functions of interest these quantities are not well-defined classically. Instead, in Definition 1, we use duality to rigorously define a semi-norm $\|f\|_{\mathcal{R}}$ that captures the essence of the above quantities and is well-defined (though possibly infinite) for any $f$ in any dimension. We show that $\|f\|_{\mathcal{R}}$ precisely captures the representational cost of $f$, and in particular is finite if and only if $f$ can be approximated arbitrarily well by a bounded norm, but possibly unbounded width, ReLU network. Our precise characterization applies to an architecture with unregularized bias terms (as in Savarese et al. (2019)) and a single unregularized linear unit— otherwise a correction accounting for a linear component is necessary, similar but more complex than the term $|f'(-\infty) + f'(+\infty)|$ in the univariate case, *i.e.*, (1).

As we uncover, the characterization of the representational cost for multivariate functions is unfortunately not as simple as the characterization (1) in the univariate case, where the Radon transform degenerates. Nevertheless, it is often easy to evaluate, and is a powerful tool for studying the representational power of bounded norm ReLU networks. Furthermore, as detailed in Section 5.5, there is no kernel function for which the associated RKHS norm is the same as (2); *i.e.*, training bounded norm neural networks is fundamentally different from kernel learning. In particular, using our characterization we show the following:

- All sufficiently smooth functions have finite representational cost, but the necessary degree of smoothness depends on the dimension. In particular, all functions in the Sobolev space $W^{d+1,1}(\mathbb{R}^d)$, *i.e.*, when all derivatives up to order $d + 1$ are $L^1$-bounded, have finite representational cost, and this cost can be bounded using the Sobolev norm. (Section 5.1)

- We calculate the representational cost of radial "bumps", and show there are bumps with finite support that have finite representational cost in all dimensions. The representational cost increases as $1/\varepsilon$ for "sharp" bumps of radius $\varepsilon$ (and fixed height). (Section 5.2)

- In dimensions greater than one, we show a general piecewise linear function with bounded support has infinite representational cost (*i.e.*, cannot be represented with a bounded norm, even with infinite networks). (Section 5.3)

- We obtain a depth separation in terms of norm: we demonstrate a function that is representable using a depth three ReLU network (*i.e.*, with two hidden layers) with small norm weights, but cannot be represented by any bounded-norm depth two (single hidden layer) ReLU network. As far as we are aware, this is the first depth separation result in terms of the norm required for representation. (Section 5.4)

**Related Work**   Although the focus of most previous work on approximation theory for neural networks was on the number of units, the norm of the weights was often used as an intermediate step. However, this use does not provide an exact characterization of the representational cost, only a (often very loose) upper bound, and in particular does not allow for depth separation results where a *lower bound* is needed. See Savarese et al. (2019) for a detailed discussion, *e.g.*, contrasting with the work of Barron (1993; 1994).

The connection between the Radon transform and two-layer neural networks was previously made by Carroll & Dickinson (1989) and Ito (1991), who used it to obtain constructive approximations when studying approximation theory in terms of network size (number of units) for threshold and sigmoidal networks. This connection also forms the foundation of ridgelet transform analysis of functions Candès & Donoho (1999); Candès (1999). More recently, Sonoda & Murata (2017) used ridgelet transform analysis to study the approximation properties of two-layer neural networks with unbounded activation functions, including the ReLU.

While working on this manuscript, we learned through discussions with Matus Telgarsky of his related parallel work. In particular, Bailey et al. (2019); Ji et al. (2019) obtained a calculation formula for the norm required to represent a radial function, paralleling our calculations in Section 5.2, and used it to show that sufficiently smooth radial functions have finite norm in any dimension, and studied how this norm changes with dimension.

## 2   INFINITE WIDTH ReLU NETWORKS

We repeat here the discussion of Savarese et al. (2019) defining the representational cost of infinite-width ReLU networks, with some corrections and changes that we highlight. Consider the collection of all two-layer networks having an unbounded number of rectified linear units (ReLUs), *i.e.*, all $g_\theta : \mathbb{R}^d \to \mathbb{R}$ defined by

$$g_\theta(\boldsymbol{x}) = \sum_{i=1}^{k} a_i [\boldsymbol{w}_i^\top \boldsymbol{x} - b_i]_+ + c, \ \text{ for all } \ \boldsymbol{x} \in \mathbb{R}^d \tag{3}$$

with parameters $\theta = (k, \boldsymbol{W} = [\boldsymbol{w}_1, ..., \boldsymbol{w}_k], \boldsymbol{b} = [b_1, ..., b_k]^\top, \boldsymbol{a} = [a_1, ..., a_k]^\top, c)$, where the width $k \in \mathbb{N}$ is unbounded. Let $\Theta$ be the collection of all such parameter vectors $\theta$. For any $\theta \in \Theta$ we let $C(\theta)$ be the sum of the squared Euclidean norm of the weights in the network excluding the bias terms, *i.e.*,

$$C(\theta) = \frac{1}{2} \left( \|\boldsymbol{W}\|_F^2 + \|\boldsymbol{a}\|^2 \right) = \frac{1}{2} \sum_{i=1}^{k} \left( \|\boldsymbol{w}_i\|_2^2 + |a_i|^2 \right), \tag{4}$$

and consider the minimal representation cost necessary to exactly represent a function $f \in \mathbb{R}^d \to \mathbb{R}$

$$R(f) := \inf_{\theta \in \Theta} C(\theta) \ \ s.t. \ \ f = g_\theta. \tag{5}$$

By the 1-homogeneity of the ReLU, it is shown in Neyshabur et al. (2014) (see also Appendix A of Savarese et al. (2019)) that minimizing $C(\theta)$ is the same as constraining the inner layer weight vectors $\{\boldsymbol{w}_i\}_{i=1}^k$ to be unit norm while minimizing the $\ell^1$-norm of the outer layer weights $\boldsymbol{a}$. Therefore, letting $\Theta'$ be the collection of all $\theta \in \Theta$ with each $\boldsymbol{w}_i$ constrained to the unit sphere $\mathbb{S}^{d-1} := \{\boldsymbol{w} \in \mathbb{R}^d : \|\boldsymbol{w}\| = 1\}$, we have

$$R(f) = \inf_{\theta \in \Theta'} \|\boldsymbol{a}\|_1 \ \ s.t. \ \ f = g_\theta. \tag{6}$$

However, we see $R(f)$ is finite only if $f$ is exactly realizable as a finite-width two layer ReLU network, *i.e.*, $f$ must be a continuous piecewise linear function with finitely many pieces. Yet, we know that any continuous function can be approximated uniformly on compact sets by allowing the number of ReLU units to grow to infinity. Since we are not concerned with the number of units, only their norm, we modify our definition of representation cost to capture this larger space of functions, and define[1]

$$\overline{R}(f) := \lim_{\varepsilon \to 0} \left( \inf_{\theta \in \Theta'} C(\theta) \ \ s.t. \ \ |g_\theta(\boldsymbol{x}) - f(\boldsymbol{x})| \le \varepsilon \ \forall \|\boldsymbol{x}\| \le 1/\varepsilon \text{ and } g_\theta(\boldsymbol{0}) = f(\boldsymbol{0}) \right) \tag{7}$$

---

[1]Our definition of $\overline{R}(f)$ differs from the one given in Savarese et al. (2019). We require $|g_\theta(\boldsymbol{x}) - f(\boldsymbol{x})| \le \varepsilon$ on the ball of radius $1/\varepsilon$ rather than all of $\mathbb{R}^d$, and we additionally require $g_\theta(\boldsymbol{0}) = f(\boldsymbol{0})$. These modifications are needed to ensure (7) and (9) are equivalent. Also, we note the choice of zero in the condition $g_\theta(\boldsymbol{0}) = f(\boldsymbol{0})$ is arbitrary and can be replaced with any point $\boldsymbol{x}_0 \in \mathbb{R}^d$.

In words, $\overline{R}(f)$ is the minimal limiting representational cost among all sequences of networks converging to $f$ uniformly (while agreeing with $f$ at zero).

Intuitively, if $\overline{R}(f)$ is finite this means $f$ is expressible as an "infinite-width" two layer ReLU network whose outer-most weights are described by a density $\alpha(\boldsymbol{w}, b)$ over all weight and bias pairs $(\boldsymbol{w}, b) \in \mathbb{S}^{d-1} \times \mathbb{R}$. To make this intuition precise, let $M(\mathbb{S}^{d-1} \times \mathbb{R})$ denote the space of signed measures $\alpha$ defined on $(\boldsymbol{w}, b) \in \mathbb{S}^{d-1} \times \mathbb{R}$ with finite total variation norm $\|\alpha\|_1 = \int_{\mathbb{S}^{d-1} \times \mathbb{R}} d|\alpha|$ (*i.e.*, the analog of the $L^1$-norm for measures), and let $c \in \mathbb{R}$. Then we define the infinite-width two-layer ReLU network $h_{\alpha,c}$ (or "infinite-width net" for short) by[2]

$$h_{\alpha,c}(\boldsymbol{x}) := \int_{\mathbb{S}^{d-1} \times \mathbb{R}} \left([\boldsymbol{w}^\top \boldsymbol{x} - b]_+ - [-b]_+\right) d\alpha(\boldsymbol{w}, b) + c \tag{8}$$

We prove in Appendix C that $\overline{R}(f)$ is equivalent to

$$\overline{R}(f) = \min_{\alpha \in M(\mathbb{S}^{d-1} \times \mathbb{R}), c \in \mathbb{R}} \|\alpha\|_1 \ \ s.t. \ \ f = h_{\alpha,c}. \tag{9}$$

Hence, learning an unbounded width ReLU network $g_\theta$ by fitting some loss functional $L(\cdot)$ while controlling the Euclidean norm of the weights $C(\theta)$ by minimizing

$$\inf_{\theta \in \Theta} L(g_\theta) + \lambda C(\theta) \tag{10}$$

is effectively the same as learning a function $f$ by controlling $\overline{R}(f)$:

$$\min_{f: \mathbb{R}^d \to \mathbb{R}} L(f) + \lambda \overline{R}(f). \tag{11}$$

In other words, $\overline{R}(f)$ captures the true inductive bias of learning with unbounded width ReLU networks having regularized weights. Our goal is then to calculate $\overline{R}(f)$ for any function $f : \mathbb{R}^d \to \mathbb{R}$, and in particular characterize when it is finite in order to understand what functions can be approximated arbitrarily well with bounded norm but unbounded width ReLU networks.

### 2.1 SIMPLIFICATION VIA UNREGULARIZED LINEAR UNIT

Every two-layer ReLU network decomposes into the sum of a network with absolute value units plus a linear part[3]. As demonstrated by Savarese et al. (2019) in the 1-D setting, the weights on the absolute value units typically determine the representational cost, with a correction term needed if the linear part has large weight. To allow for a cleaner formulation of the representation cost without this correction term, we consider adding in *one additional unregularized linear unit* $\boldsymbol{v}^\top \boldsymbol{x}$ (similar to a "skip connection") to "absorb" any representational cost due to the linear part.

Namely, for any $\theta \in \Theta$ and $\boldsymbol{v} \in \mathbb{R}^d$ we define the class of unbounded with two-layer ReLU networks $g_{\theta,\boldsymbol{v}}$ with a linear unit by $g_{\theta,\boldsymbol{v}}(\boldsymbol{x}) = g_\theta(\boldsymbol{x}) + \boldsymbol{v}^\top \boldsymbol{x}$ where $g_\theta$ is as defined in (3), and associate $g_{\theta,\boldsymbol{v}}$ with the same weight norm $C(\theta)$ as defined in (4) (*i.e.*, we exclude the norm of the weight $\boldsymbol{v}$ on the additional linear unit from the cost). We then define the representational cost $\overline{R}_1(f)$ for this class of networks by

$$\overline{R}_1(f) := \lim_{\varepsilon \to 0} \left( \inf_{\theta \in \Theta', \boldsymbol{v} \in \mathbb{R}^d} C(\theta) \ \ s.t. \ \ |g_{\theta,\boldsymbol{v}}(\boldsymbol{x}) - f(\boldsymbol{x})| \leq \varepsilon \ \forall \|\boldsymbol{x}\| \leq 1/\varepsilon \text{ and } g_\theta(\boldsymbol{0}) = f(\boldsymbol{0}) \right). \tag{12}$$

Likewise, for all $\alpha \in M(\mathbb{S}^{d-1} \times \mathbb{R})$, $\boldsymbol{v} \in \mathbb{R}^d$, $c \in \mathbb{R}$, we define an infinite width net with a linear unit by $h_{\alpha,\boldsymbol{v},c}(\boldsymbol{x}) := h_{\alpha,c}(\boldsymbol{x}) + \boldsymbol{v}^\top \boldsymbol{x}$. We prove in Appendix C that $\overline{R}_1(f)$ is equivalent to:

$$\overline{R}_1(f) = \min_{\alpha \in M(\mathbb{S}^{d-1} \times \mathbb{R}), \boldsymbol{v} \in \mathbb{R}^d, c \in R} \|\alpha\|_1 \ \ s.t. \ \ f = h_{\alpha,\boldsymbol{v},c}. \tag{13}$$

In fact, we show the minimizer of (13) is unique and is characterized as follows:

---

[2]Our definition of $h_{\alpha,c}$ also differs from the one given in Savarese et al. (2019). To ensure the integral is well-defined, we include the additional $-[-b]_+$ term in the integrand. See Remark 1 in Appendix B for more discussion on this point.

[3]Such a decomposition follows immediately from the identity $[t]_+ = \frac{1}{2}(|t| + t)$

**Lemma 1.** $\overline{R}_1(f) = \|\alpha^+\|_1$ where $\alpha^+ \in M(\mathbb{S}^{d-1} \times \mathbb{R})$ is the unique even measure[4] such that $f = h_{\alpha^+, \boldsymbol{v}, c}$ for some $\boldsymbol{v} \in \mathbb{R}^d$, $c \in \mathbb{R}$.

The proof of Lemma 1 is given in Appendix D. The uniqueness in Lemma 1 allows for a more explicit characterization $\overline{R}_1(f)$ in function space relative to $\overline{R}(f)$, as we show in Section 4.

## 3 THE RADON TRANSFORM AND ITS DUAL

Our characterization of the representational cost in Section 4 is posed in terms of *the Radon transform* — a transform that is fundamental to computational imaging, and whose inverse is the basis of image reconstruction in computed tomography. For an investigation of its properties and applications, see Helgason (1999). Here we give a brief review of the Radon transform and its dual as needed for subsequent derivations; readers familiar with these topics can skip to Section 4.

The Radon transform $\mathcal{R}$ represents a function $f : \mathbb{R}^d \to \mathbb{R}$ in terms of its integrals over all possible hyperplanes in $\mathbb{R}^d$, as parameterized by the unit normal direction to the hyperplane $\boldsymbol{w} \in \mathbb{S}^{d-1}$ and the signed distance of the hyperplane from the origin $b \in \mathbb{R}$:

$$\mathcal{R}\{f\}(\boldsymbol{w}, b) := \int_{\boldsymbol{w}^\top \boldsymbol{x} = b} f(\boldsymbol{x}) \, ds(\boldsymbol{x}) \text{ for all } (\boldsymbol{w}, b) \in \mathbb{S}^{d-1} \times \mathbb{R}, \tag{14}$$

where $ds(\boldsymbol{x})$ represents integration with respect to $(d-1)$-dimensional surface measure on the hyperplane $\boldsymbol{w}^\top \boldsymbol{x} = b$. Note the Radon transform is an *even* function, *i.e.*, $\mathcal{R}\{f\}(\boldsymbol{w}, b) = \mathcal{R}\{f\}(-\boldsymbol{w}, -b)$ for all $(\boldsymbol{w}, b) \in \mathbb{S}^{d-1} \times \mathbb{R}$, since the equations $\boldsymbol{w}^\top \boldsymbol{x} = b$ and $-\boldsymbol{w}^\top \boldsymbol{x} = -b$ determine the same hyperplane. See Figure 1 for an illustration of the Radon transform in dimension $d = 2$.

The Radon transform is invertible for many common spaces of functions, and its inverse is a composition of the *dual Radon transform* $\mathcal{R}^*$ (*i.e.*, the adjoint of $\mathcal{R}$) followed by a filtering step in Fourier domain. The dual Radon transform $\mathcal{R}^*$ maps a function $\varphi : \mathbb{S}^{d-1} \times \mathbb{R} \to \mathbb{R}$ to a function over $\boldsymbol{x} \in \mathbb{R}^d$ by integrating over the subset of coordinates $(\boldsymbol{w}, b) \in \mathbb{S}^{d-1} \times \mathbb{R}$ corresponding to all hyperplanes passing through $\boldsymbol{x}$:

$$\mathcal{R}^*\{\varphi\}(\boldsymbol{x}) := \int_{\mathbb{S}^{d-1}} \varphi(\boldsymbol{w}, \boldsymbol{w}^\top \boldsymbol{x}) \, d\boldsymbol{w} \text{ for all } \boldsymbol{x} \in \mathbb{R}^d \tag{15}$$

where $d\boldsymbol{w}$ represents integration with respect to the surface measure of the unit sphere $\mathbb{S}^{d-1}$. The filtering step is given by a $(d-1)/2$-power of the (negative) Laplacian $(-\Delta)^{(d-1)/2}$, where for any $s > 0$ the operator $(-\Delta)^{s/2}$ is defined in Fourier domain by

$$\widehat{(-\Delta)^{s/2} f}(\boldsymbol{\xi}) = \|\boldsymbol{\xi}\|^s \widehat{f}(\boldsymbol{\xi}), \tag{16}$$

using $\widehat{g}(\boldsymbol{\xi}) := (2\pi)^{-d/2} \int g(\boldsymbol{x}) e^{-i\boldsymbol{\xi}^\top \boldsymbol{x}} d\boldsymbol{x}$ to denote the $d$-dimensional Fourier transform at the Fourier domain (frequency) variable $\boldsymbol{\xi} \in \mathbb{R}^d$. When $d$ is odd, $(-\Delta)^{(d-1)/2}$ is the same as applying the usual Laplacian $(d-1)/2$ times, *i.e.*, $(-\Delta)^{(d-1)/2} = (-1)^{(d-1)/2} \Delta^{(d-1)/2}$, while if $d$ is even it is a pseudo-differential operator given by convolution with a singular kernel. Combining these two operators gives the inversion formula $f = \gamma_d (-\Delta)^{(d-1)/2} \mathcal{R}^*\{\mathcal{R}\{f\}\}$, where $\gamma_d$ is a constant depending on dimension $d$, which holds for $f$ belonging to many common function spaces (see, *e.g.*, Helgason (1999)).

The dual Radon transform is also invertible by a similar formula, albeit under more restrictive conditions on the function space. We use the following formula due to Solmon (1987) that holds for all Schwartz class functions[5] on $\mathbb{S}^{d-1} \times \mathbb{R}$, which we denote by $\mathcal{S}(\mathbb{S}^{d-1} \times \mathbb{R})$:

**Lemma 2** (Solmon (1987)). *If $\varphi$ is an even function[6], i.e., $\varphi(-\boldsymbol{w}, -b) = \varphi(\boldsymbol{w}, b)$ for all $(\boldsymbol{w}, b) \in \mathbb{S}^{d-1} \times \mathbb{R}$, belonging to the Schwartz class $\mathcal{S}(\mathbb{S}^{d-1} \times \mathbb{R})$, then*

$$\gamma_d \mathcal{R}\{(-\Delta)^{(d-1)/2} \mathcal{R}^*\{\varphi\}\} = \varphi, \tag{17}$$

*where $\gamma_d = \frac{1}{2(2\pi)^{d-1}}$.*

---

[4]Roughly speaking, a measure $\alpha$ is even if $\alpha(\boldsymbol{w}, b) = \alpha(-\boldsymbol{w}, -b)$ for all $(\boldsymbol{w}, b) \in \mathbb{S}^{d-1} \times \mathbb{R}$; see Appendix B for a precise definition.

[5]*i.e.*, functions $\varphi : \mathbb{S}^{d-1} \times \mathbb{R} \to \mathbb{R}$ that are $C^\infty$-smooth such that $\varphi(\boldsymbol{w}, b)$ and all its partial derivatives decrease faster than $O(|b|^{-N})$ as $|b| \to \infty$ for any $N \geq 0$

[6]The assumption that $\varphi$ is even is necessary since odd functions are annihilated by $\mathcal{R}^*$.

## 4 REPRESENTATIONAL COST IN FUNCTION SPACE: THE $\mathcal{R}$-NORM

Our starting point is to relate the Laplacian of an infinite-width net to the dual Radon transform of its defining measure. In particular, consider an infinite width net $f$ defined in terms of a smooth density $\alpha(\boldsymbol{w}, b)$ over $\mathbb{S}^{d-1} \times \mathbb{R}$ that decreases rapidly to zero with $|b| \to \infty$, so that we can write

$$f(\boldsymbol{x}) = \int_{\mathbb{S}^{d-1} \times \mathbb{R}} \left([\boldsymbol{w}^\top \boldsymbol{x} - b]_+ - [-b]_+\right) \alpha(\boldsymbol{w}, b)\, d\boldsymbol{w}\, db + \boldsymbol{v}^\top \boldsymbol{x} + c. \tag{18}$$

Differentiating twice inside the integral, the Laplacian $\Delta f(\boldsymbol{x}) = \sum_{i=1}^{d} \partial_{x_i}^2 f(\boldsymbol{x})$ is given by

$$\Delta f(\boldsymbol{x}) = \int_{\mathbb{S}^{d-1} \times \mathbb{R}} \delta(\boldsymbol{w}^\top \boldsymbol{x} - b) \alpha(\boldsymbol{w}, b)\, d\boldsymbol{w}\, db = \int_{\mathbb{S}^{d-1}} \alpha(\boldsymbol{w}, \boldsymbol{w}^\top \boldsymbol{x})\, d\boldsymbol{w}. \tag{19}$$

where $\delta(\cdot)$ denotes a Dirac delta[7]. We see that the right-hand side of (19) is precisely the dual Radon transform of $\alpha$, *i.e.*, we have shown $\Delta f = \mathcal{R}^*\{\alpha\}$. Applying the inversion formula for the dual Radon transform given in (17) to this identity, and using the characterization of $\overline{R}_1(f)$ given in Lemma 1, immediately gives the following result.

**Lemma 3.** *Suppose $f = h_{\alpha, \boldsymbol{v}, c}$ for some $\alpha \in \mathcal{S}\left(\mathbb{S}^{d-1} \times \mathbb{R}\right)$ with $\alpha$ even, and $\boldsymbol{v} \in \mathbb{R}^d$, $c \in \mathbb{R}$. Then $\alpha = -\gamma_d \mathcal{R}\{(-\Delta)^{(d+1)/2} f\}$, and $\overline{R}_1(f) = \gamma_d \|\mathcal{R}\{(-\Delta)^{(d+1)/2} f\}\|_1$ where $\gamma_d = \frac{1}{2(2\pi)^{d-1}}$.*

This result suggests that more generally if we are given any function $f$, we ought to be able to compute $\overline{R}_1(f)$ using the formula in Lemma 3. The following result, proved in Appendix D, shows this is indeed the case assuming $f$ is integrable and sufficiently smooth, which for simplicity we state in the case of odd dimensions $d$.[8].

**Proposition 1.** *Suppose $d$ is odd. If both $f \in L^1(\mathbb{R}^d)$ and $\Delta^{(d+1)/2} f \in L^1(\mathbb{R}^d)$, then*

$$\overline{R}_1(f) = \gamma_d \|\mathcal{R}\{\Delta^{(d+1)/2} f\}\|_1 = \gamma_d \|\partial_b^{d+1} \mathcal{R}\{f\}\|_1 < \infty. \tag{20}$$

Here we used the intertwining property of the Radon transform and the Laplacian to write $\mathcal{R}\{\Delta^{(d+1)/2} f\} = \partial_b^{d+1} \mathcal{R}\{f\}$ (see Appendix A for more details).

Given these results, one might expect for an *arbitrary* function $f$ we should have $\overline{R}_1(f)$ equal to one of the expressions in (20). However, for many functions of interest these quantities are not classically well-defined. For example, the finite-width ReLU net $f(\boldsymbol{x}) = \sum_{i=1}^{n} a_i [\boldsymbol{w}_i^\top \boldsymbol{x} - b_i]_+$ is a piecewise linear function that is non-smooth along each hyperplane $\boldsymbol{w}_i^\top \boldsymbol{x} = b_i$, so its derivatives can only be understood in the distributional sense. Similarly, in this case the Radon transform of $f$ is not well-defined since $f$ is unbounded and not integrable along hyperplanes.

Instead, we use duality to define a functional (the "$\mathcal{R}$-norm") that extends to the more general case where $f$ is possibly non-smooth or not integrable along hyperplanes. In particular, we define a functional on the space of all Lipschitz continuous functions[9]. The main idea is to re-express the $L^1$-norm in (20) as a supremum of the inner product over a space of dual functions $\psi : \mathbb{S}^{d-1} \times \mathbb{R} \to \mathbb{R}$, *i.e.*, using the fact $\mathcal{R}^*$ is the adjoint of $\mathcal{R}$ and the Laplacian $\Delta$ is self-adjoint we write

$$\|\mathcal{R}\{\Delta^{(d+1)/2} f\}\|_1 = \sup_{\|\psi\|_\infty \le 1} \langle \mathcal{R}\{\Delta^{(d+1)/2} f\}, \psi \rangle = \sup_{\|\psi\|_\infty \le 1} \langle f, \Delta^{(d+1)/2} \mathcal{R}^*\{\psi\} \rangle \tag{21}$$

then restrict $\psi$ to a space where $\Delta^{(d+1)/2} \mathcal{R}^*\{\psi\}$ is always well-defined. More formally, we have:

**Definition 1.** *For any Lipschitz continuous function $f : \mathbb{R}^d \to \mathbb{R}$ define its $\mathcal{R}$-norm[10] $\|f\|_{\mathcal{R}}$ by*

$$\|f\|_{\mathcal{R}} := \sup\left\{-\gamma_d \langle f, (-\Delta)^{(d+1)/2} \mathcal{R}^*\{\psi\}\rangle : \psi \in \mathcal{S}(\mathbb{S}^{d-1} \times \mathbb{R}), \psi \text{ even}, \|\psi\|_\infty \le 1\right\}. \tag{22}$$

*where $\gamma_d = \frac{1}{2(2\pi)^{d-1}}$, $\mathcal{S}(\mathbb{S}^{d-1} \times \mathbb{R})$ is the space of Schwartz functions on $\mathbb{S}^{d-1} \times \mathbb{R}$, and $\langle f, g \rangle := \int_{\mathbb{R}^d} f(\boldsymbol{x}) g(\boldsymbol{x}) d\boldsymbol{x}$. If $f$ is not Lipschitz we define $\|f\|_{\mathcal{R}} = +\infty$.*

---

[7] Here we use Dirac deltas informally; for a formal derivation of (19) see Lemma 9 in Appendix D.

[8] For $d$ even, Proposition 1 holds with the pseudo-differential operators $(-\Delta)^{(d+1)/2}$ and $(-\partial_b^2)^{(d+1)/2}$ in place of $\Delta^{(d+1)/2}$ and $\partial_b^{d+1}$; see Section 3.

[9] Recall that $f$ is Lipschitz continuous if there exists a constant $L$ (depending on $f$) such that $|f(\boldsymbol{x}) - f(\boldsymbol{y})| \le L\|\boldsymbol{x} - \boldsymbol{y}\|$ for all $\boldsymbol{x}, \boldsymbol{y} \in \mathbb{R}^d$.

[10] Strictly speaking, the functional $\|\cdot\|_{\mathcal{R}}$ is not a norm, but it is a semi-norm on the space of functions for which it is finite; see Appendix F.

We prove in Appendix D that the $\mathcal{R}$-norm is well-defined, though not always finite, for all Lipschitz functions and, whether finite or infinite, it is always equal to the representational cost $\overline{R}_1(\cdot)$:

**Theorem 1.** $\overline{R}_1(f) = \|f\|_\mathcal{R}$ for all functions $f$. In particular, $\overline{R}_1(f)$ is finite if and only if $f$ is Lipschitz and $\|f\|_\mathcal{R}$ is finite.

We give the proof of Theorem 1 in Appendix D, but the following example illustrates many of its key elements.

**Example 1.** *We compute $\overline{R}_1(f) = \|f\|_\mathcal{R}$ in the case where $f$ is a finite-width two-layer ReLU network. First, consider the case where $f$ consists of a single ReLU unit: $f(\boldsymbol{x}) = a_1[\boldsymbol{w}_1^\top \boldsymbol{x} - b_1]_+$ for some $a_1 \in \mathbb{R}$ and $(\boldsymbol{w}_1, b_1) \in \mathbb{S}^{d-1}$. Note that $\Delta f(\boldsymbol{x}) = a\,\delta(\boldsymbol{w}_1^\top \boldsymbol{x} - b_1)$ in a distributional sense, i.e., for any smooth test function $\varphi$ we have $\langle \Delta f, \varphi \rangle = \langle f, \Delta \varphi \rangle = a_1 \int \varphi(\boldsymbol{x})\delta(\boldsymbol{w}_1^\top \boldsymbol{x} - b_1)d\boldsymbol{x} = a_1 \mathcal{R}\{\varphi\}(\boldsymbol{w}_1, b_1)$. So for any even $\psi \in \mathcal{S}(\mathbb{S}^{d-1} \times \mathbb{R})$ we have*

$$-\gamma_d \langle f, (-\Delta)^{(d+1)/2} \mathcal{R}^* \{\psi\} \rangle = \gamma_d \langle \Delta f, (-\Delta)^{(d-1)/2} \mathcal{R}^* \{\psi\} \rangle \tag{23}$$

$$= a_1 \gamma_d \mathcal{R}\{(-\Delta)^{(d-1)/2} \mathcal{R}^* \{\psi\}\}(\boldsymbol{w}_1, b_1) \tag{24}$$

$$= a_1 \psi(\boldsymbol{w}_1, b_1) \tag{25}$$

*where in the last step we used the inversion formula (17). Since the supremum defining $\|f\|_\mathcal{R}$ is over all even $\psi \in \mathcal{S}(\mathbb{S}^{d-1} \times \mathbb{R})$ such that $\|\psi\|_\infty \leq 1$, taking any $\psi^*$ such that $\psi^*(\boldsymbol{w}_1, b_1) = sign(a_1)$ and $|\psi^*(\boldsymbol{w}_1, b_1)| \leq 1$ otherwise, we see that $\|f\|_\mathcal{R} = |a_1|$. The general case now follows by linearity: let $f(\boldsymbol{x}) = \sum_{i=1}^k a_i [\boldsymbol{w}_i^\top \boldsymbol{x} - b_i]_+$ such that all the pairs $\{(\boldsymbol{w}_i, b_i)\}_{i=1}^k \cup \{(-\boldsymbol{w}_i, -b_i)\}_{i=1}^k$ are distinct. Then for any $\psi \in \mathcal{S}(\mathbb{S}^{d-1} \times \mathbb{R})$ we have*

$$-\gamma_d \langle f, (-\Delta)^{(d+1)/2} \mathcal{R}^* \{\psi\} \rangle = \sum_{i=1}^k a_i \psi(\boldsymbol{w}_i, b_i). \tag{26}$$

*Letting $\psi^*$ be any even Schwartz function such that $\psi^*(\boldsymbol{w}_i, b_i) = \psi^*(-\boldsymbol{w}_i, -b_i) = sign(a_i)$ for all $i = 1, ..., k$ and $|\psi^*(\boldsymbol{w}, b)| \leq 1$ otherwise, we see that $\overline{R}_1(f) = \|f\|_\mathcal{R} = \sum_{i=1}^k |a_i|$.*

The representational cost $\overline{R}(f)$ defined without the unregularized linear unit is more difficult to characterize explicitly. However, we prove that $\overline{R}(f)$ is finite if and only if $\|f\|_\mathcal{R}$ is finite, and give bounds for $\overline{R}(f)$ in terms of $\|f\|_\mathcal{R}$ and the norm of the gradient of the function "at infinity", similar to the expressions derived in Savarese et al. (2019) in the 1-D setting.

**Theorem 2.** $\overline{R}(f)$ *is finite if and only if $\|f\|_\mathcal{R}$ is finite, in which case we have the bounds*

$$\max\{\|f\|_\mathcal{R}, 2\|\nabla f(\infty)\|\} \leq \overline{R}(f) \leq \|f\|_\mathcal{R} + 2\|\nabla f(\infty)\|, \tag{27}$$

*where $\nabla f(\infty) := \lim_{r \to \infty} \frac{1}{c_d r^{d-1}} \oint_{\|\boldsymbol{x}\|=r} \nabla f(\boldsymbol{x}) ds(\boldsymbol{x}) \in \mathbb{R}^d$ with $c_d := \int_{\mathbb{S}^{d-1}} d\boldsymbol{w} = \frac{2\pi^{d/2}}{\Gamma(d/2)}$. In particular, if $\nabla f(\infty) = \boldsymbol{0}$ then $\overline{R}(f) = \overline{R}_1(f) = \|f\|_\mathcal{R}$.*

We give the proof of Theorem 2 in Appendix E. The lower bound $\max\{\|f\|_\mathcal{R}, 2\|\nabla f(\infty)\|\}$ is analogous to the expression for the 1D representational cost (1) obtained in Savarese et al. (2019). From this, one might speculate that $\overline{R}(f)$ is equal to $\max\{\|f\|_\mathcal{R}, 2\|\nabla f(\infty)\|_2\}$. However, in Appendix E we show this is not the case: there are examples of functions $f$ in all dimensions such that $\overline{R}(f)$ attains the upper bound in a non-trivial way (*e.g.*, $f(x, y) = |x| + y$ in $d = 2$).

### 4.1 PROPERTIES OF THE $\mathcal{R}$-NORM

In Appendix F we prove several useful properties for the $\mathcal{R}$-norm. In particular, we show the $\mathcal{R}$-norm is in fact a *semi-norm*, *i.e.*, it is absolutely homogeneous and satisfies the triangle inequality, while $\|f\|_\mathcal{R} = 0$ if and only if $f$ is affine. We also show the $\mathcal{R}$-norm is invariant to coordinate translation and rotations, and prove the following scaling law under contractions/dilation:

**Proposition 2.** *If $f_\varepsilon(\boldsymbol{x}) := f(\boldsymbol{x}/\varepsilon)$ for any $\varepsilon > 0$, then $\|f_\varepsilon\|_\mathcal{R} = \varepsilon^{-1} \|f\|_\mathcal{R}$*

Proposition 2 shows that "spikey" functions will necessarily have large $\mathcal{R}$-norm. For example, let $f$ be any non-negative function supported on the ball of radius 1 with maximum height 1 such that $\|f\|_\mathcal{R}$ is finite. Then the contraction $f_\varepsilon$ is supported on the ball of radius $\varepsilon$ with maximum height

1, but $\|f_\varepsilon\|_\mathcal{R} = \varepsilon^{-1} \|f\|_\mathcal{R}$ blows up as $\varepsilon \to 0$. From a generalization perspective, the fact that the $\mathcal{R}$-norm blows up with contractions is a desirable property, since otherwise the minimum norm fit to data would be spikes on data points. In particular, this is what would happen if the representational cost involved derivatives lower than $d + 1$, and so in this sense it is not a coincidence that $\|f\|_\mathcal{R}$ involves derivatives of order $d + 1$; for more discussion on this point see Section 5.1.

Finally, we also show that having finite $\mathcal{R}$-norm implies strong conditions in Fourier domain. In particular, we show that if the $\mathcal{R}$-norm of an $L^1$ function is finite then its Fourier transform must decay rapidly along every ray. A precise statement is given in Proposition 12 in Appendix F.

## 5 Consequences, Applications and Discussion

Our characterization of the representational cost for multivariate functions in terms of the $\mathcal{R}$-norm is unfortunately not as simple as the characterization in the univariate case. Nevertheless, it is often easy to evaluate, and is a powerful tool for studying the representational power of bounded norm ReLU networks.

### 5.1 Sobolev spaces

Here we relate Sobolev spaces and the $\mathcal{R}$-norm. The key result is the following upper bound, which is proved in Appendix G.

**Proposition 3.** *If* $f : \mathbb{R}^d \to \mathbb{R}$ *is Lipschitz and* $(-\Delta)^{(d+1)/2} f$ *exists in a weak sense*[11] *then*

$$\|f\|_\mathcal{R} \leq c_d \gamma_d \|(-\Delta)^{(d+1)/2} f\|_1. \tag{28}$$

*where* $c_d = \int_{\mathbb{S}^{d-1}} d\boldsymbol{w} = \frac{2\pi^{d/2}}{\Gamma(d/2)}$, *and* $\gamma_d = \frac{1}{2(2\pi)^{d-1}}$.

Recall that if the dimension $d$ is odd then $(-\Delta)^{(d+1)/2}$ is just an integer power of the negative Laplacian, which is a linear combination of partial derivatives of order $d + 1$. Hence, we have $\|(-\Delta)^{(d+1)/2} f\|_1 \leq c_d \gamma_d \|f\|_{W^{d+1,1}}$, where $\|f\|_{W^{d+1,1}}$ is the Sobolev norm given by the sum of the $L^1$-norm of $f$ and the $L^1$-norms of all its weak partial derivatives up to order $d + 1$. This gives the following immediate corollary to Proposition 3:

**Corollary 1.** *Suppose* $d$ *is odd. If* $f$ *belongs to the Sobolev space* $W^{d+1,1}(\mathbb{R}^d)$, *i.e.,* $f$ *and all its weak derivatives up to order* $d + 1$ *are in* $L^1(\mathbb{R}^d)$, *then* $\|f\|_\mathcal{R}$ *is finite and* $\|f\|_\mathcal{R} \leq c_d \gamma_d \|f\|_{W^{d+1,1}}$.

The $(d + 1)$-order smoothness in Corollary 1 is optimal in the sense that the $\mathcal{R}$-norm cannot be uniformly bounded by a $L^1$-type Sobolev norm with smoothness of a lower order. To see this, let $D^s$ represent any $s$-order partial derivative, and let $f$ be any compactly supported function such that $\|D^s f\|_1$ and $\|f\|_\mathcal{R}$ are finite. Then the contraction $f_\varepsilon(\boldsymbol{x}) := f(\boldsymbol{x}/\varepsilon)$ obeys the scaling law $\|D^s f_\varepsilon\|_1 = \varepsilon^{d-s} \|D^s f\|_1$ and so $\|f_\varepsilon\|_{W^{s,1}} = O(\varepsilon^{d-s})$ as $\varepsilon \to 0$. Also, by Proposition 2 we have $\|f_\varepsilon\|_\mathcal{R} = \varepsilon^{-1} \|f\|_\mathcal{R}$. So the only way $\|f_\varepsilon\|_\mathcal{R}$ could be uniformly bounded by $\|f_\varepsilon\|_{W^{s,1}}$ as $\varepsilon \to 0$ is if $s \geq d + 1$.

The smoothness requirements in Corollary 1 are also necessary from a generalization perspective. In particular, if the representational cost were bounded by $L^1$-norm of derivatives of order strictly less than $d$, we could not expect generalization from finite data. This is because the data could be fit exactly with a sum of spike functions whose representational cost could be made arbitrarily small by shrinking the support of the spikes.

Here it is also interesting to compare with (Bach, 2017, Proposition 5). There it is shown that the space of all two-layer infinite-width ReLU networks defined by $L^2$-bounded densities contains a Sobolev space $W^{s,2}$ with $s > (d + 1)/2$. The $s > d/2$ scaling of smoothness is necessary when considering an $L^2$-type Sobolev norm, since by a similar argument to the above, it is the scaling that ensures contractions $f_\varepsilon$ have unbounded norm as $\varepsilon \to 0$.

---

[11]*i.e., for all compactly supported smooth functions* $\varphi$ *there exists a locally integrable function* $g \in L^1_{\text{loc}}(\mathbb{R}^d)$ *such that* $\int f (-\Delta)^{(d+1)/2} \varphi \, d\boldsymbol{x} = \int g\varphi \, d\boldsymbol{x}$.

## 5.2 RADIAL BUMP FUNCTIONS

To help build intuition with the $\mathcal{R}$-norm and illustrate its scaling with dimension, here we study the case where $f$ is a radially symmetric function, *i.e.*, $f(\boldsymbol{x}) = g(\|\boldsymbol{x}\|)$ for some function $g : [0, \infty) \to \mathbb{R}$. In this case, the $\mathcal{R}$-norm is expressible entirely in terms of derivatives of the radial profile function $g$, as shown in the following result, which is proved in Appendix H.

**Proposition 4.** *Suppose $d \geq 3$ is odd. If $f \in L^1(\mathbb{R}^d)$ with $f(\boldsymbol{x}) = g(\|\boldsymbol{x}\|)$ then*

$$\|f\|_{\mathcal{R}} = \frac{2}{(d-2)!} \int_0^\infty \left| \partial^{(d+1)} \rho(b) \right| db \ \text{ where } \ \rho(b) := \int_b^\infty g(t)(t^2 - b^2)^{(d-3)/2} t \, dt. \quad (29)$$

For example, in the $d = 3$ dimensional case, we have

$$\|f\|_{\mathcal{R}} = 2 \int_0^\infty |b \, \partial^3 g(b) + 3\partial^2 g(b)| db, \quad (d = 3) \quad (30)$$

More generally, for any odd dimension $d \geq 3$ a simple induction shows (29) is equivalent to

$$\|f\|_{\mathcal{R}} = \frac{2}{(d-2)!} \int_0^\infty |Q_d\{g\}(b)| db \quad (31)$$

where $Q_d$ is a differential operator of degree $(d+3)/2$ having the form $Q_d = \sum_{k=2}^{(d+3)/2} p_{k,d}(b)\partial^k$ where each $p_{k,d}(b)$ is a polynomial in $b$ of degree $k-2$. In particular, if the weak derivative $\partial^{(d+1)/2}g$ exists and has bounded variation, then $\|f\|_{\mathcal{R}}$ is finite.

**Example 2.** *Suppose $d \geq 3$ is odd. Consider the radial bump function $f_{d,k}(\boldsymbol{x}) = g_{d,k}(\|\boldsymbol{x}\|)$ with $\boldsymbol{x} \in \mathbb{R}^d$ where for any integer $k > 0$ we define*

$$g_{d,k}(r) = \begin{cases} (1 - r^2)^k & \text{if } 0 \leq r < 1 \\ 0 & \text{if } r \geq 1, \end{cases} \quad (32)$$

*We prove $\|f_{d,k}\|_{\mathcal{R}}$ is finite if and only if $k \geq \frac{d+1}{2}$ (see Appendix H). To illustrate the scaling with dimension $d$, in Appendix H we also prove that for the choice $k_d = (d+1)/2 + 2$ we have the bounds $(d+5)d \leq \|f_{d,k_d}\|_{\mathcal{R}} \leq 2d(d+5)$, hence $\|f_{d,k_d}\|_{\mathcal{R}} \sim d^2$. Similarly, by the dilation property (2), a contraction of $f_{d,k_d}$ to the ball of radius $\varepsilon$ has $\mathcal{R}$-norm scaling as $\sim d^2/\varepsilon$.*

The next example[12] shows there there is a universal choice of radial bump function in all (odd) dimensions with finite $\mathcal{R}$-norm:

**Example 3.** *Suppose $d \geq 3$ is odd. Consider the radial bump function $f(\boldsymbol{x}) = g(\|\boldsymbol{x}\|)$ with $\boldsymbol{x} \in \mathbb{R}^d$ where*

$$g(r) = \begin{cases} e^{-\frac{1}{1-r^2}} & \text{if } 0 \leq r < 1 \\ 0 & \text{if } r \geq 1. \end{cases} \quad (33)$$

*Since $g$ is $C^\infty$-smooth and its derivatives of all orders are $L^1$-bounded, $f$ has finite $\mathcal{R}$-norm by Proposition 4.*

## 5.3 PIECEWISE LINEAR FUNCTIONS

Every finite-width two-layer ReLU network is a continuous piecewise linear function. However, the reverse is not true. For example, in dimensions two and above no compactly supported piecewise linear function is expressible as a finite-width two-layer ReLU network. A natural question then is: what piecewise linear functions are represented by bounded norm infinite-width nets, *i.e.*, have finite $\mathcal{R}$-norm? In particular, do compactly supported piecewise linear functions have finite $\mathcal{R}$-norm? Here we show this is generally not the case.

**Proposition 5.** *Suppose $f : \mathbb{R}^d \to \mathbb{R}$ is a continuous piecewise linear function with compact support, where the boundary sets between regions satisfy additional mild conditions. Then $f$ has infinite $\mathcal{R}$-norm.*

---

[12]The existence of such a radial function was noted in parallel work by Matus Telgarsky. Discussions with Telgarsky motivated us to construct and analyze it using the $\mathcal{R}$-norm.

See Appendix I for proof and for the precise conditions needed on the boundary sets. Roughly speaking, the result holds for any "generic" piecewise linear function with compact support, *i.e.*, if a function does not satisfy these conditions, then some small perturbation of the function does.

This result suggests that the space of piecewise linear functions expressible as a bounded norm infinite-width two-layer ReLU network is not qualitatively different the space of finite-width networks. We go further and make the following conjecture:

**Conjecture 1.** *A continuous piecewise linear function $f$ has finite $\mathcal{R}$-norm if and only if it is exactly representable by a finite-width two-layer ReLU network.*

### 5.4 DEPTH SEPARATION

In an effort to understand the power of deeper networks, there has been much work showing how some functions can be much more easily approximated *in terms of number of required units* by deeper networks compared to shallower ones, including results showing how functions that can be well-approximated by three-layer networks require a much larger number of units to approximate if using a two-layer network (e.g. Pinkus (1999); Telgarsky (2016); Liang & Srikant (2016); Safran & Shamir (2017); Yarotsky (2017)). The following example shows that, also in terms of the norm, such a depth separation exists for ReLU nets:

**Example 4.** *The pyramid function $f(\boldsymbol{x}) = [1 - \|\boldsymbol{x}\|_1]_+$ is a compactly supported piecewise linear function satisfying the conditions needed for Proposition 5 to hold[13], hence has infinite representational cost as a two-layer ReLU network ($\overline{R}(f) = \overline{R}_1(f) = +\infty$), but can be exactly represented as a finite-width three-layer ReLU network.*

Interestingly, this result shows that, in terms of the norm, we have a qualitative rather then quantitative depth separation: the minimal norm of the weights needed to represent the function with three layers is finite, while with only two layers it is not merely very large, but *infinite*. In contrast, in standard depth separation results, the separation is quantitative: we can compensate for a decrease in depth and use more neurons to achieve the same approximation quality. It would be interesting to further strengthen Example 4 by obtaining a quantitative lower bound on the norm required to $\epsilon$-approximate the pyramid with an infinite-width two-layer ReLU network.

### 5.5 THE $\mathcal{R}$-NORM IS NOT A RKHS NORM

There is an ongoing debate in the community on whether neural network learning can be simulated or replicated by kernel machines with the "right" kernel. In this context, it is interesting to ask whether the inductive bias we uncover can be captured by a kernel, or in other words, whether the $\mathcal{R}$-norm is an RKHS-norm[14] The answer is no:

**Proposition 6.** *The $\mathcal{R}$-norm is not an RKHS norm.*

This is seen immediately by the failure of the parallelogram law to hold. For example, if $f_1(\boldsymbol{x}) = [\boldsymbol{w}_1^\top \boldsymbol{x}]_+$, $f_2 = [\boldsymbol{w}_2^\top \boldsymbol{x}]_+$ with $\boldsymbol{w}_1, \boldsymbol{w}_2 \in \mathbb{S}^{d-1}$ distinct, then by Example 1 we have $\|f_1\|_{\mathcal{R}} = \|f_2\|_{\mathcal{R}} = 1$, while $\|f_1 + f_2\|_{\mathcal{R}} = \|f_1 - f_2\|_{\mathcal{R}} = 2$, and so $2(\|f_1\|_{\mathcal{R}}^2 + \|f_2\|_{\mathcal{R}}^2) \neq \|f_1 + f_2\|_{\mathcal{R}}^2 + \|f_1 - f_2\|_{\mathcal{R}}^2$.

### 5.6 GENERALIZATION IMPLICATIONS

Neyshabur et al. (2015) shows that training an unbounded-width neural network while regularizing the $\ell_2$ norm of the weights results in a sample complexity proportional to a variant[15] of $\overline{R}(f)$. This paper gives an explicit characterization of $\overline{R}(f)$ and thus of the sample complexity of learning a function using regularized unbounded-width neural networks.

---

[13]More precisely, it satisfies condition (b) of Proposition 16 in Appendix I.

[14]To be precise, the $\mathcal{R}$-norm is only a semi-norm on the space of functions for which it is finite, but it becomes a norm on the quotient space obtained by modding out by all functions with zero $\mathcal{R}$-norm, *i.e.*, all affine functions. The question is whether the $\mathcal{R}$-norm on this quotient space is an RKHS norm.

[15]Their analysis does not allow for unregularized bias, but can be extended to allow for it.

ACKNOWLEDGMENTS

We are grateful to Matus Telgarsky (University of Illinois, Urbana-Champaign) for stimulating discussions, including discussing his yet unpublished work with us. In particular, Telgarsky helped us refine our view of radial bumps and realize a fixed radial function can have finite norm in all dimensions. We would also like to thank Guillaume Bal (University of Chicago) for helpful discussions regarding the Radon transform, and Jason Altschuler (MIT) for pointers regarding convergence of measures and Prokhorov's Theorem. Some of the work was done while DS and NS were visiting the Simons Institute for Theoretical Computer Science as participants in the Foundations of Deep Learning Program. NS was partially supported by NSF awards 1764032 and 1546500. DS was partially supported by the Israel Science Foundation (grant No. 31/1031), and by the Taub Foundation. RW and GO were partially supported by AFOSR award FA9550-18-1-0166, DOE award DE-AC02-06CH11357, and NSF awards OAC-1934637 and DMS-1930049.

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

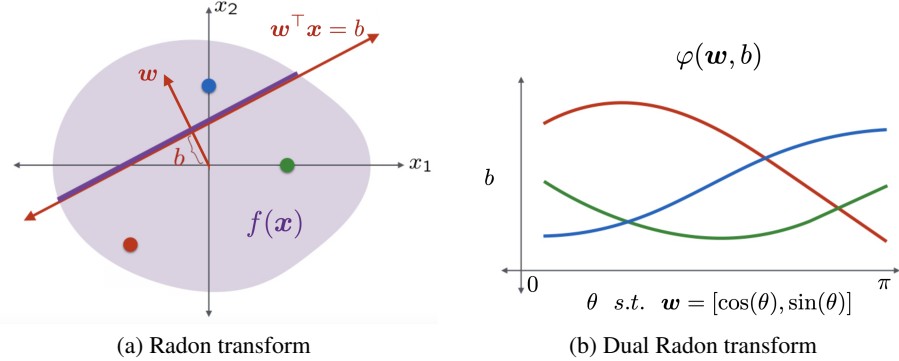

(a) Radon transform

(b) Dual Radon transform

Figure 1: Radon transform. (a) Illustration of the Radon transform in equation (14) in dimension $d = 2$. The red line of points $x$ satisfying $w^\top x = b$ defines the domain of the integral over $f(x)$, where $w$ determines the line orientation (angle relative to the coordinate axes) and $b$ determines its offset from the origin. (b) Illustration of the support of the Radon transform for $f(x) = \delta(x - (-1, -1))$ (red), $f(x) = \delta(x - (1, 0))$ (green), and $f(x) = \delta(x - (0, 1))$ (blue). If a function $f$ is a superposition of such $\delta$ functions, then $\mathcal{R}\{f\}$ is the sum of the curves in (b); this is typically referred to as a "sinogram". Furthermore, the dual Radon transform in equation (15) integrates any function $\varphi(w, b)$ over all curves like one of the three in (b).

## APPENDICES

## A   ADDITIONAL PROPERTIES OF THE RADON TRANSFORM

Figure 1 illustrates the Radon transform and its dual in dimension $d = 2$.

We will often use the fact that the Radon transform is a bounded linear operator from $L^1(\mathbb{R}^d)$ to $L^1(\mathbb{S}^{d-1} \times \mathbb{R})$, *i.e.*, if $f \in L^1(\mathbb{R}^d)$ then $\mathcal{R}\{f\} \in L^1(\mathbb{R}^d)$. In particular, if $f \in L^1(\mathbb{R}^d)$ then $\mathcal{R}\{f\}$ is defined almost everywhere on $\mathbb{S}^{d-1} \times \mathbb{R}$, and the function $\mathcal{R}\{f\}(w, \cdot)$ is in $L^1(\mathbb{R})$ for all $w \in \mathbb{S}^{d-1}$.

Here we also recall the *Fourier slice theorem* for the Radon transform (see, e.g., Helgason (1999)): Let $f \in L^1(\mathbb{R}^d)$, then for all $\sigma \in \mathbb{R}$ and $w \in \mathbb{S}^{d-1}$ we have

$$\mathcal{F}_b \mathcal{R}\{f\}(w, \sigma) = \widehat{f}(\sigma \cdot w) \tag{34}$$

where $\mathcal{F}_b$ indicates the 1-D Fourier transform in the offset variable $b$. From this it is easy to establish the following *intertwining* property of the Laplacian and the Radon transform: assuming $f$ and $\Delta f$ are in $L^1(\mathbb{R}^d)$, we have

$$\mathcal{R}\{\Delta f\} = \partial_b^2 \mathcal{R}\{f\} \tag{35}$$

where $\partial_b$ is the partial derivative in the offset variable $b$. More generally for any positive integer $s$, assuming $f$ and $(-\Delta)^{s/2} f$ are in $L^1(\mathbb{R}^d)$ we have

$$\mathcal{R}\{(-\Delta)^{s/2} f\} = (-\partial_b^2)^{s/2} \mathcal{R}\{f\} \tag{36}$$

where fractional powers of $-\partial_b^2$ can be defined in Fourier domain, same as fractional powers of the Laplacian. In particular, if $d$ is odd, $(-\partial_b^2)^{(d+1)/2} = (-1)^{(d+1)/2} \partial_b^{d+1}$, while if $d$ is even, $(-\partial_b^2)^{(d+1)/2} = (\mathcal{H}\partial_b)^{d+1}$ where $\mathcal{H}$ is the Hilbert transform in the offset variable $b$.

## B   INFINITE-WIDTH NETS

**Measures and infinite-width nets**   Let $\alpha$ be a signed measure [16] defined on $\mathbb{S}^{d-1} \times \mathbb{R}$, and let $\|\alpha\|_1 = \int d|\alpha|$ denote its total variation norm. We let $M(\mathbb{S}^{d-1} \times \mathbb{R})$ denote the space of measures on $\mathbb{S}^{d-1} \times \mathbb{R}$ with finite total variation norm. Since $\mathbb{S}^{d-1} \times \mathbb{R}$ is a locally compact space, $M(\mathbb{S}^{d-1} \times \mathbb{R})$

---

[16] To be precise, we assume $\alpha$ is a signed *Radon* measure; see, e.g., Malliavin (2012) for a formal definition. We omit the word "Radon" and simply call $\alpha$ a measure to avoid confusion with the Radon transform, which is central to this work.

is the Banach space dual of $C_0(\mathbb{S}^{d-1} \times \mathbb{R})$, the space of continuous functions on $\mathbb{S}^{d-1} \times \mathbb{R}$ vanishing at infinity (Malliavin, 2012, Chapter 2, Theorem 6.6), and

$$\|\alpha\|_1 = \sup \left\{ \int \varphi \, d\alpha : \varphi \in C_0(\mathbb{S}^{d-1} \times \mathbb{R}), \|\varphi\|_\infty \le 1 \right\}. \tag{37}$$

For any $\alpha \in M(\mathbb{S}^{d-1} \times \mathbb{R})$ and $\varphi \in C_0(\mathbb{S}^{d-1} \times \mathbb{R})$, we often use $\langle \alpha, \varphi \rangle$ to denote $\int \varphi d\alpha$.

Any $\alpha \in M(\mathbb{S}^{d-1} \times \mathbb{R})$ can be extended uniquely to a continuous linear functional on $C_b(\mathbb{S}^{d-1} \times \mathbb{R})$, the space continuous and bounded functions on $\mathbb{S}^{d-1} \times \mathbb{R}$. In particular, since the function $\varphi(\boldsymbol{w}, b) = [\boldsymbol{w}^\top \boldsymbol{x} - b]_+ - [-b]_+$ belongs to $C_b(\mathbb{S}^{d-1} \times \mathbb{R})$, we see that the infinite-width net

$$h_\alpha(\boldsymbol{x}) := \int_{\mathbb{S}^{d-1} \times \mathbb{R}} ([\boldsymbol{w}^\top \boldsymbol{x} - b]_+ - [-b]_+) d\alpha(\boldsymbol{w}, b) \tag{38}$$

is well-defined for all $\boldsymbol{x} \in \mathbb{R}^d$.

**Remark 1.** Our definition of an infinite-width net in differs slightly from Savarese et al. (2019): we integrate a constant shift of the ReLU $[\boldsymbol{w}^\top \boldsymbol{x} - b]_+ - [-b]_+$ with respect to the measure $\alpha(\boldsymbol{w}, b)$ rather than $[\boldsymbol{w}^\top \boldsymbol{x} - b]_+$ as in Savarese et al. (2019). As shown above, this ensures the integral is always well-defined for any measure $\alpha$ with finite total variation. Alternatively, we could have restricted to measures that have finite first moment, *i.e.*, $\int_{\mathbb{S}^{d-1} \times \mathbb{R}} |b| \, d|\alpha|(\boldsymbol{w}, b) < \infty$, which ensures the definition $\widetilde{h}_\alpha(\boldsymbol{x}) := \int_{\mathbb{S}^{d-1} \times \mathbb{R}} [\boldsymbol{w}^\top \boldsymbol{x} - b]_+ d\alpha(\boldsymbol{w}, b)$ proposed in Savarese et al. (2019) is always well-defined. However, restricting to measures with finite first moment complicates the function space description, and excludes from our analysis certain functions that are still naturally defined as limits of bounded norm finite-width networks, and so we opt for the definition above instead. In the case that $\alpha$ has a finite first moment the difference between definitions is immaterial since $h_\alpha$ and $\widetilde{h}_\alpha$ are equal up to an additive constant, which implies they have the same representational cost under $\overline{R}(\cdot)$ and $\overline{R}_1(\cdot)$.

**Even and odd measures**    We say $\alpha \in M(\mathbb{S}^{d-1} \times \mathbb{R})$ is *even* if

$$\int_{\mathbb{S}^{d-1} \times \mathbb{R}} \varphi(\boldsymbol{w}, b) d\alpha(\boldsymbol{w}, b) = \int_{\mathbb{S}^{d-1} \times \mathbb{R}} \varphi(-\boldsymbol{w}, -b) d\alpha(\boldsymbol{w}, b) \text{ for all } \varphi \in C_0(\mathbb{S}^{d-1} \times \mathbb{R}) \tag{39}$$

or $\alpha$ is *odd* if

$$\int_{\mathbb{S}^{d-1} \times \mathbb{R}} \varphi(\boldsymbol{w}, b) d\alpha(\boldsymbol{w}, b) = - \int_{\mathbb{S}^{d-1} \times \mathbb{R}} \varphi(-\boldsymbol{w}, -b) d\alpha(\boldsymbol{w}, b) \text{ for all } \varphi \in C_0(\mathbb{S}^{d-1} \times \mathbb{R}). \tag{40}$$

It is easy to show every measure $\alpha \in M(\mathbb{S}^{d-1} \times \mathbb{R})$ is uniquely decomposable as $\alpha = \alpha^+ + \alpha^-$ where $\alpha^+$ is even and $\alpha^-$ is odd, which we call the even and odd decomposition of $\alpha$. For example, if $\alpha$ has a density $\mu(\boldsymbol{w}, b)$ then $\alpha^+$ is the measure with density $\mu^+(\boldsymbol{w}, b) = \frac{1}{2}(\mu(\boldsymbol{w}, b) + \mu(-\boldsymbol{w}, -b))$ and $\alpha^-$ is the measure with density $\mu^-(\boldsymbol{w}, b) = \frac{1}{2}(\mu(\boldsymbol{w}, b) - \mu(-\boldsymbol{w}, -b))$.

We let $M(\mathbb{P}^d)$ denote the subspace of all even measures in $M(\mathbb{S}^{d-1} \times \mathbb{R})$, which is the Banach space dual of $C_0(\mathbb{P}^d)$, the subspace of all even functions $\varphi \in C_0(\mathbb{S}^{d-1} \times \mathbb{R})$. Even measures play an important role in our results because of the following observations.

Let $\alpha \in M(\mathbb{S}^{d-1} \times \mathbb{R})$ with even and odd decomposition $\alpha = \alpha^+ + \alpha^-$. Then we have $h_\alpha = h_{\alpha^+} + h_{\alpha^-}$. By the identity $[t]_+ + [-t]_+ = |t|$ we can show

$$h_{\alpha^+}(\boldsymbol{x}) = \frac{1}{2} \int_{\mathbb{S}^{d-1} \times \mathbb{R}} (|\boldsymbol{w}^\top \boldsymbol{x} + b| - |b|) d\alpha^+(\boldsymbol{w}, b). \tag{41}$$

Likewise, by the identity $[t]_+ - [-t]_+ = t$ we have

$$h_{\alpha^-}(\boldsymbol{x}) = \boldsymbol{v}_0^\top \boldsymbol{x}. \tag{42}$$

where $\boldsymbol{v}_0 = \frac{1}{2} \int_{\mathbb{S}^{d-1} \times \mathbb{R}} \boldsymbol{w} d\alpha^-(\boldsymbol{w}, b)$. Hence, $h_\alpha$ decomposes into a sum of a component with absolute value activations and a linear function. In particular, if $f = h_{\alpha, \boldsymbol{v}, c}$ for some $\alpha \in M(\mathbb{S}^{d-1} \times \mathbb{R}), \boldsymbol{v} \in \mathbb{R}^d, c \in \mathbb{R}$, letting $\alpha^+$ be the even part of $\alpha$, we always have $f = h_{\alpha^+, \boldsymbol{v}', c}$ for some $\boldsymbol{v}' \in \mathbb{R}^d$. In other words, we lose no generality by restricting ourselves to infinite width nets of the form $f = h_{\alpha, \boldsymbol{v}, c}$ where $\alpha$ is even (*i.e.*, $\alpha \in M(\mathbb{P}^d)$).

We will need the following fact about even and odd decompositions of measures under the total variation norm:

**Proposition 7.** *Let $\alpha \in M(\mathbb{S}^{d-1} \times \mathbb{R})$ with $\alpha = \alpha^+ + \alpha^-$ where $\alpha^+$ is even and $\alpha^-$ is odd. Then $\|\alpha^+\|_1 \leq \|\alpha\|_1$ and $\|\alpha^-\|_1 \leq \|\alpha\|_1$.*

*Proof.* For any $\varphi \in C_0(\mathbb{S}^{d-1} \times \mathbb{R})$ we can write $\varphi = \varphi_+ + \varphi_-$ where $\varphi_+(\boldsymbol{w}, b) = \frac{1}{2}(\varphi(\boldsymbol{w}, b) + \varphi(-\boldsymbol{w}, -b))$ is even and $\varphi_-(\boldsymbol{w}, b) = \frac{1}{2}(\varphi(\boldsymbol{w}, b) - \varphi(-\boldsymbol{w}, -b))$ is odd. Note that $\int \varphi \, d\alpha^+ = \int \varphi_+ \, d\alpha^+$ since $\int \varphi_- \, d\alpha^+ = 0$. Furthermore, if $|\varphi(\boldsymbol{w}, b)| \leq 1$ for all $(\boldsymbol{w}, b) \in \mathbb{S}^{d-1} \times \mathbb{R}$ we see that $|\varphi_+(\boldsymbol{w}, b)| \leq \frac{1}{2}(|\varphi(\boldsymbol{w}, b)| + |\varphi(-\boldsymbol{w}, -b)|) \leq 1$ for all $(\boldsymbol{w}, b) \in \mathbb{S}^{d-1} \times \mathbb{R}$. Therefore, in the dual definition of $\|\alpha^+\|_1$ given in (37) it suffices to take the supremum over all even functions $\varphi \in C_0(\mathbb{S}^{d-1} \times \mathbb{R})$. Hence,

$$\|\alpha\|_1 = \sup\left\{\int \varphi \, d\alpha : \varphi \in C_0(\mathbb{S}^{d-1} \times \mathbb{R}), \|\varphi\|_\infty \leq 1\right\} \tag{43}$$

$$= \sup\left\{\int \varphi \, d\alpha^+ + \int \varphi \, d\alpha^- : \varphi \in C_0(\mathbb{S}^{d-1} \times \mathbb{R}), \|\varphi\|_\infty \leq 1\right\} \tag{44}$$

$$\geq \sup\left\{\int \varphi \, d\alpha^+ + \int \varphi \, d\alpha^- : \varphi \in C_0(\mathbb{S}^{d-1} \times \mathbb{R}), \|\varphi\|_\infty \leq 1, \varphi \text{ even}\right\} \tag{45}$$

$$= \sup\left\{\int \varphi \, d\alpha^+ : \varphi \in C_0(\mathbb{S}^{d-1} \times \mathbb{R}), \|\varphi\|_\infty \leq 1, \varphi \text{ even}\right\} \tag{46}$$

$$= \|\alpha^+\|_1 \tag{47}$$

A similar argument shows $\|\alpha^-\|_1 \leq \|\alpha\|_1$. $\qquad\square$

**Lipschitz continuity of infinite-width nets** Define $\mathrm{Lip}(\mathbb{R}^d)$ to be the space of all real-valued Lipschitz continuous functions on $\mathbb{R}^d$. For any $f \in \mathrm{Lip}(\mathbb{R}^d)$, define $\|f\|_L := \sup_{\boldsymbol{x} \neq \boldsymbol{y}} \frac{|f(\boldsymbol{x}) - f(\boldsymbol{y})|}{\|\boldsymbol{x} - \boldsymbol{y}\|}$, *i.e.*, the smallest possible Lipschitz constant. The following result shows that $\mathrm{Lip}(\mathbb{R}^d)$ is a natural space to work in when considering infinite-width nets:

**Proposition 8** (Infinite-width nets are Lipschitz). *Let $f = h_{\alpha, \boldsymbol{v}, c}$ for any $\alpha \in M(\mathbb{S}^{d-1} \times \mathbb{R}), \boldsymbol{v} \in \mathbb{R}^d, c \in \mathbb{R}$. Then $f \in \mathrm{Lip}(\mathbb{R}^d)$ with $\|f\|_L \leq \|\alpha\|_1 + \|\boldsymbol{v}\|$.*

*Proof.* First we prove for all even $\alpha \in M(\mathbb{P}^d)$, $\|h_\alpha\|_L \leq \|\alpha\|_1/2$.

By the reverse triangle inequality we have $\left||\boldsymbol{w}^\top \boldsymbol{x} - b| - |\boldsymbol{w}^\top \boldsymbol{y} - b|\right| \leq \left|\boldsymbol{w}^\top(\boldsymbol{x} - \boldsymbol{y})\right|$ for all $\boldsymbol{x}, \boldsymbol{y} \in \mathbb{R}^d, \boldsymbol{w} \in \mathbb{S}^{d-1}, b \in \mathbb{R}$. Therefore, using identity (41), for all $\boldsymbol{x}, \boldsymbol{y} \in \mathbb{R}^d$ we see that

$$|h_\alpha(\boldsymbol{x}) - h_\alpha(\boldsymbol{y})| = \frac{1}{2}\left|\int_{\mathbb{S}^{d-1} \times \mathbb{R}} \left(|\boldsymbol{w}^\top \boldsymbol{x} - b| - |\boldsymbol{w}^\top \boldsymbol{y} - b|\right) d\alpha(\boldsymbol{w}, b)\right| \tag{48}$$

$$\leq \frac{1}{2}\int_{\mathbb{S}^{d-1} \times \mathbb{R}} \left||\boldsymbol{w}^\top \boldsymbol{x} - b| - |\boldsymbol{w}^\top \boldsymbol{y} - b|\right| d|\alpha|(\boldsymbol{w}, b) \tag{49}$$

$$\leq \frac{1}{2}\int_{\mathbb{S}^{d-1} \times \mathbb{R}} |\boldsymbol{w}^\top(\boldsymbol{x} - \boldsymbol{y})| d|\alpha|(\boldsymbol{w}, b) \tag{50}$$

$$\leq \frac{1}{2}\|\boldsymbol{x} - \boldsymbol{y}\|\|\alpha\|_1 \tag{51}$$

which shows $h_\alpha$ is Lipschitz with $\|h_\alpha\|_L \leq \|\alpha\|_1/2$.

More generally, for any infinite-width net $f = h_{\alpha, \boldsymbol{v}, c}$ with $\alpha \in M(\mathbb{S}^{d-1} \times \mathbb{R})$, $\boldsymbol{v} \in \mathbb{R}^d$ and $c \in \mathbb{R}$. From the even and odd decomposition $\alpha = \alpha^+ + \alpha^-$ we have $f = h_{\alpha^+, \boldsymbol{v}_0 + \boldsymbol{v}, c}$, where $\boldsymbol{v}_0 = \frac{1}{2}\int_{\mathbb{S}^{d-1} \times \mathbb{R}} \boldsymbol{w} d\alpha^-(\boldsymbol{w}, b)$. Hence, $\|\boldsymbol{v}_0\|_2 \leq \|\alpha^-\|_1/2$, Therefore, by the triangle inequality, $\|f\|_L \leq \|\alpha^+\|_1/2 + \|\alpha^-\|_1/2 + \|\boldsymbol{v}\| \leq \|\alpha\|_1 + \|\boldsymbol{v}\|$, which gives the claim. $\qquad\square$

## C  Optimization characterization of representational cost

Here we establish the optimization equivalents of the representational costs $\overline{R}(f)$ and $\overline{R}_1(f)$ given in (9) and (13).

As an intermediate step, we first give equivalent expressions for $\overline{R}(f)$ and $\overline{R}_1(f)$ in terms of sequences finite-width two-layer ReLU networks converging pointwise to $f$. For this we need to introduce some additional notation and definitions.

We let $\mathcal{A}(\mathbb{S}^{d-1} \times \mathbb{R})$ denote the space of all measures given by a finite linear combination of Diracs, *i.e.*, all $\alpha \in M(\mathbb{S}^{d-1} \times \mathbb{R})$ of the form $\alpha = \sum_{i=1}^{k} a_i \delta_{(\boldsymbol{w}_i, b_i)}$ for some $a_i \in \mathbb{R}$, $(\boldsymbol{w}_i, b_i) \in \mathbb{S}^{d-1} \times \mathbb{R}$, $i = 1, ..., k$, where $\delta_{(\boldsymbol{w}, b)}$ denotes a Dirac delta at location $(\boldsymbol{w}, b) \in \mathbb{S}^{d-1} \times \mathbb{R}$. We call any $\alpha \in \mathcal{A}(\mathbb{S}^{d-1} \times \mathbb{R})$ a *discrete* measure.

Note there is a one-to-one correspondence between discrete measures and finite-width two layer ReLU nets (up to a bias term). Namely, for any $\theta \in \Theta'$ defining a finite-width net $g_\theta(\boldsymbol{x}) = \sum_{i=1}^{k} a_i [\boldsymbol{w}_i^\top \boldsymbol{x} - b_i]_+ + c$, setting $\alpha = \sum_{i=1}^{k} a_i \delta_{(\boldsymbol{w}_i, b_i)}$ we have $f = h_{\alpha, c'}$ with $c' = g_\theta(\boldsymbol{0})$. We write $\theta \in \Theta' \leftrightarrow \alpha \in \mathcal{A}(\mathbb{S}^{d-1} \times \mathbb{R})$ to indicate this correspondence. Furthermore, in this case $C(\theta) = \sum_{i=1}^{k} |a_i| = \|\alpha\|_1$.

We also recall some facts related to the convergence of sequences of measures. Let $C_b(\mathbb{S}^{d-1} \times \mathbb{R})$ denote the set of all continuous and bounded functions on $\mathbb{S}^{d-1} \times \mathbb{R}$. A sequence of measures $\{\alpha_n\}$, with $\alpha_n \in M(\mathbb{S}^{d-1} \times \mathbb{R})$ is said to converge *narrowly* to a measure $\alpha \in M(\mathbb{S}^{d-1} \times \mathbb{R})$ if $\int \varphi \, d\alpha_n \to \int \varphi \, d\alpha$ for all $\varphi \in C_b(\mathbb{S}^{d-1} \times \mathbb{R})$. Also, a sequence $\{\alpha_n\}$ is called *tight* if for all $\varepsilon > 0$ there exists a compact set $K_\varepsilon \subset \mathbb{S}^{d-1} \times \mathbb{R}$ such that $|\alpha_n|(K_\varepsilon^c) \leq \varepsilon$ for all $n$ sufficiently large. Every narrowly convergent sequence of measures is tight (Malliavin, 2012, Theorem 6.8). Conversely, any sequence $\{\alpha_n\}$ that is tight and uniformly bounded in total variation norm has a narrowly convergent subsequence; this is due to a version of Prokhorov's Theorem for signed measures (Bogachev, 2007, Theorem 8.6.2).

Now we establish the following equivalent expressions for the representational costs $\overline{R}(\cdot)$ and $\overline{R}_1(\cdot)$.

**Lemma 4.** *For any $f : \mathbb{R}^d \to \mathbb{R}$ let $f_0$ denote the function $f_0(\boldsymbol{x}) = f(\boldsymbol{x}) - f(\boldsymbol{0})$. For $\overline{R}(f)$ as defined in (7) and $\overline{R}_1(f)$ as defined in (12), we have*

$$\overline{R}(f) = \inf \left\{ \limsup_{n \to \infty} \|\alpha_n\|_1 : \alpha_n \in \mathcal{A}(\mathbb{S}^{d-1} \times \mathbb{R}), \ h_{\alpha_n} \to f_0 \text{ pointwise}, \ \{\alpha_n\} \text{ tight} \right\}. \quad (52)$$

*and*

$$\overline{R}_1(f) = \inf \left\{ \limsup_{n \to \infty} \|\alpha_n\|_1 : \alpha_n \in \mathcal{A}(\mathbb{S}^{d-1} \times \mathbb{R}), \boldsymbol{v}_n \in \mathbb{R}^d, \ h_{\alpha_n, \boldsymbol{v}_n, 0} \to f_0 \text{ pointwise}, \ \{\alpha_n\} \text{ tight} \right\}. \quad (53)$$

*Proof.* We prove the identity in (52) for $\overline{R}(f)$; the identity in (53) for $\overline{R}_1(f)$ follows by the same argument. Define

$$R_\varepsilon(f) := \inf_{\theta \in \Theta'} C(\theta) \ \ s.t. \ \ |g_\theta(\boldsymbol{x}) - f(\boldsymbol{x})| \leq \varepsilon \ \forall \ \|\boldsymbol{x}\| \leq 1/\varepsilon \text{ and } g_\theta(\boldsymbol{0}) = f(\boldsymbol{0}) \quad (54)$$

so that $\overline{R}(f) = \lim_{\varepsilon \to 0} R_\varepsilon(f)$. Also, let $L(f)$ denote the right-hand side of (52).

First, suppose $\overline{R}(f)$ is finite. Let $\varepsilon_n = 1/n$. Then by definition of $\overline{R}(f)$, for all $n$ there exists $\theta_n \in \Theta'$ such that $C(\theta_n) \leq R_{\varepsilon_n}(f) + \varepsilon_n$, while $|g_{\theta_n}(\boldsymbol{x}) - f(\boldsymbol{x})| \leq \varepsilon_n$ for $\|\boldsymbol{x}\| \leq 1/\varepsilon_n$ and $g_{\theta_n}(\boldsymbol{0}) = f(\boldsymbol{0})$. Note that $\theta_n \in \Theta' \leftrightarrow \alpha_n \in M(\mathbb{S}^{d-1} \times \mathbb{R})$ with $g_{\theta_n} = h_{\alpha_n, c}$ where $c = g_{\theta_n}(\boldsymbol{0}) = f(\boldsymbol{0})$ and $\|\alpha_n\|_1 = C(\theta_n)$. Hence, $h_{\alpha_n}(\boldsymbol{x}) = g_{\theta_n}(\boldsymbol{x}) - f(\boldsymbol{0})$, and we have $|h_{\alpha_n}(\boldsymbol{x}) - f_0(\boldsymbol{x})| = |g_{\theta_n}(\boldsymbol{x}) - f(\boldsymbol{x})| \leq \varepsilon_n$ for $\|\boldsymbol{x}\| \leq 1/\varepsilon_n$. Therefore, $h_{\alpha_n} \to f_0$ pointwise, while

$$\limsup_{n \to \infty} \|\alpha_n\|_1 \leq \limsup_{n \to \infty} (R_{\varepsilon_n}(f) + \varepsilon_n) = \overline{R}(f), \quad (55)$$

which shows $L(f) \leq \overline{R}(f)$. Finally, it suffices to show $\{\alpha_n\}$ has a tight subsequence, since we can reproduce the steps above with respect to the subsequence. Towards this end, define $q_n(\boldsymbol{x}) = \int |\boldsymbol{w}^\top \boldsymbol{x} - b| d|\alpha_n|(\boldsymbol{w}, b)$, which is well-defined since $\alpha_n$ is discrete and has compact support. Then $q_n$ is Lipschitz with $\|q_n\|_L \leq \|\alpha_n\|_1 \leq B$ for some finite $B$, hence the sequence $\{q_n\}$ is uniformly Lipschitz. By the Arzela-Ascoli Theorem, $\{q_n\}$ has a subsequence $\{q_{n_k}\}$ that converges uniformly on compact subsets. In particular, $q_{n_k}(\boldsymbol{0}) = \int |b| d|\alpha_{n_k}|(\boldsymbol{w}, b) \leq L < \infty$ for some $L$, which implies the sequence $\{\alpha_{n_k}\}$ is tight.

Conversely, suppose $L(f)$ is finite. Fix any $\varepsilon > 0$. Then by definition of $L(f)$ there exists a sequence $\alpha_n \in M(\mathbb{S}^{d-1} \times \mathbb{R}) \leftrightarrow \theta_n \in \Theta'$ such that $\lim_{n \to \infty} \|\alpha_n\|_1$ exists with $\lim_{n \to \infty} \|\alpha_n\|_1 < L(f) + \varepsilon$, while $f_n := h_{\alpha_n,c} = g_{\theta_n}$ with $c = f(\mathbf{0})$ converges to $f$ pointwise and satisfies $f_n(\mathbf{0}) = f(\mathbf{0})$ for all $n$. Since, $\lim_{n \to \infty} \|\alpha_n\|_1 < L(f) + \varepsilon$, there exists an $N_1$ such that for all $n \geq N_1$ we have $\|\alpha_n\|_1 \leq L(f) + \varepsilon$. By Proposition 8, the Lipschitz constant of $f_n$ is bounded above by $\|\alpha_n\|_1$ for all $n$, hence the sequence $f_n$ is uniformly Lipschitz. This implies $f_n \to f$ uniformly on compact subsets, and so there exists an $N_2$ such that $\|f_n(\mathbf{x}) - f(\mathbf{x})\| \leq \varepsilon$ for all $\|\mathbf{x}\| \leq 1/\varepsilon$ and $f_n(\mathbf{0}) = f(\mathbf{0})$ for all $n \geq N_2$. For all $n \geq N_2$, $f_n$ satisfies the constraints in the definition of $R_\varepsilon(\cdot)$. Therefore, for all $n \geq \max\{N_1, N_2\}$ we have

$$R_\varepsilon(f) \leq C(\theta_n) = \|\alpha_n\|_1 \leq L(f) + \varepsilon. \tag{56}$$

Taking the limit as $\varepsilon \to 0$, we get $\overline{R}(f) \leq L(f)$. Therefore, we have shown $\overline{R}(f)$ is finite if and only if $L(f)$ is finite, in which case $\overline{R}(f) = L(f)$, giving the claim. $\qquad\square$

The following lemma shows every infinite-width net is the pointwise limit of a sequence of finite-width nets defined in terms of sequence of measures uniformly bounded in total variation norm.

**Lemma 5.** *Let $f = h_{\alpha,\mathbf{v},c}$ for any $\alpha \in M(\mathbb{S}^{d-1} \times \mathbb{R}), \mathbf{v} \in \mathbb{R}^d$, and $c \in \mathbb{R}$. Then there exists a sequence of discrete measures $\alpha_n \in \mathcal{A}(\mathbb{S}^{d-1} \times \mathbb{R})$ with $\|\alpha_n\|_1 \leq \|\alpha\|_1$ such that $f_n = h_{\alpha_n,\mathbf{v},c}$ converges to $f$ pointwise.*

*Proof.* For any $\alpha \in M(\mathbb{S}^{d-1} \times \mathbb{R})$ there exists a sequence of discrete measures $\alpha_n$ converging narrowly to $\alpha$ such that $\|\alpha_n\|_1 \leq \|\alpha\|_1$ (Malliavin, 2012, Chapter 2, Theorem 6.9). Let $f_n = h_{\alpha_n,\mathbf{v},c}$. Since the function $(\mathbf{w}, b) \mapsto [\mathbf{w}^\top \mathbf{x} - b]_+ - [-b]_+$ is continuous and bounded, we have $f_n(\mathbf{x}) \to f(\mathbf{x})$ for all $\mathbf{x} \in \mathbb{R}^d$, i.e., $f_n \to f$ pointwise. $\qquad\square$

**Lemma 6.** *We have the equivalences*

$$\overline{R}(f) = \min_{\alpha \in M(\mathbb{S}^{d-1} \times \mathbb{R}), c \in \mathbb{R}} \|\alpha\|_1 \;\; s.t. \;\; f = h_{\alpha,c}, \tag{57}$$

*and*

$$\overline{R}_1(f) = \min_{\alpha \in M(\mathbb{S}^{d-1} \times \mathbb{R}), \mathbf{v} \in \mathbb{R}^d, c \in \mathbb{R}} \|\alpha\|_1 \;\; s.t. \;\; f = h_{\alpha,\mathbf{v},c}. \tag{58}$$

*Proof.* We prove the $\overline{R}(f)$ case; the $\overline{R}_1(f)$ case follows by the same argument. Throughout the proof we use the equivalence of $\overline{R}(f)$ given in Lemma 4, and let $\mathcal{M}(f)$ denote the right-hand side of (57).

Assume $\overline{R}(f)$ is finite. Then there exists a tight sequence $\{\alpha_n\}$, $\alpha_n \in \mathcal{A}(\mathbb{S}^{d-1} \times \mathbb{R})$, that is uniformly bounded in total variation norm such that $h_{\alpha_n} \to f_0$ pointwise. Therefore, by Prokhokov's Theorem, $\{\alpha_n\}$ has a subsequence $\{\alpha_{n_k}\}$ converging narrowly to a measure $\alpha$, hence $f_0 = h_\alpha$. Furthermore, narrow convergence implies $\|\alpha\|_1 \leq \limsup_{k \to \infty} \|\alpha_{n_k}\|_1 \leq \limsup_{n \to \infty} \|\alpha_n\|_1$, and so $\mathcal{M}(f) \leq \limsup_{n \to \infty} \|\alpha_n\|_1$. Taking the infimum over all such sequences $\{\alpha_n\}$, we have $\mathcal{M}(f) \leq \overline{R}(f)$.

Conversely, assume $\mathcal{M}(f)$ is finite. Let $\alpha \in M(\mathbb{S}^{d-1} \times \mathbb{R})$ be any measure such that $f_0 = h_\alpha$. By Lemma 5 there exists a sequence $\{\alpha_n\}$, $\alpha_n \in \mathcal{A}(\mathbb{S}^{d-1} \times \mathbb{R})$, such that $h_{\alpha_n} \to f_0$ pointwise, while $\|\alpha_n\|_1 \leq \|\alpha\|_1$. Hence, $\overline{R}(f) \leq \limsup_{n \to \infty} \|\alpha_n\|_1 \leq \|\alpha\|_1$. Since this holds for any $\alpha$ with $f_0 = h_\alpha$, we see that $\overline{R}(f) \leq \mathcal{M}(f)$, proving the claim. $\qquad\square$

Now we show that if $f$ is an infinite-width net, $\overline{R}_1(f)$ is equal to the minimal total variation norm of all even measures defining $f$ (in fact, later we show for every infinite-width net is defined in terms of a *unique* even measure, whose total variation norm is equal to $\overline{R}_1(f)$; see Lemma 10).

**Lemma 7.** *We have*

$$\overline{R}_1(f) = \min_{\alpha^+ \in M(\mathbb{P}^d), \mathbf{v} \in \mathbb{R}^d, c \in \mathbb{R}} \|\alpha^+\|_1 \;\; s.t. \;\; f = h_{\alpha^+,\mathbf{v},c}. \tag{59}$$

*where the minimization is over all even $\alpha^+ \in M(\mathbb{P}^d)$.*

*Proof.* Suppose $f = h_{\alpha,\boldsymbol{v},c}$ for some $\alpha \in M(\mathbb{S}^{d-1} \times \mathbb{R}), \boldsymbol{v} \in \mathbb{R}^d, c \in \mathbb{R}$. If $\alpha$ has even and odd decomposition $\alpha = \alpha^+ + \alpha^-$ then $f = h_{\alpha^+,\boldsymbol{0},0} + h_{\alpha^-,\boldsymbol{v},c} = h_{\alpha^+,\boldsymbol{v}',c}$ for some $\boldsymbol{v}' \in \mathbb{R}^d$. Also, by Proposition 7, we have $\|\alpha^+\|_1 \le \|\alpha^+ + \alpha^-\|_1 = \|\alpha\|_1$ for any $\alpha^-$ odd. Hence, the optimization problem describing $\overline{R}_1(f)$ in (58) reduces to (59). $\qquad\square$

## D  EXTENSION OF $\mathcal{R}$-NORM TO LIPSCHITZ FUNCTIONS AND PROOF OF THEOREM 1

To simplify notation we let $\mathcal{S}(\mathbb{P}^d)$ denote the space of *even* Schwartz functions on $\mathbb{S}^{d-1}\times\mathbb{R}$, *i.e.*, $\psi \in \mathcal{S}(\mathbb{P}^d)$ if $\psi \in \mathcal{S}(\mathbb{S}^{d-1} \times \mathbb{R})$ with $\psi(\boldsymbol{w},b) = \psi(-\boldsymbol{w},-b)$ for all $(\boldsymbol{w},b) \in \mathbb{S}^{d-1} \times \mathbb{R}$.

We will need a finer characterization of the image of Schwartz functions under the dual Radon transform than what is given in Lemma 9, which is also due to Solmon (1987):

**Lemma 8** (Solmon (1987), Theorem 7.7). *Let $\psi \in \mathcal{S}(\mathbb{P}^d)$ and define $\varphi = \gamma_d(-\Delta)^{(d-1)/2}\mathcal{R}^*\{\psi\}$. Then $\varphi \in C^\infty(\mathbb{R}^d)$ with $\varphi(\boldsymbol{x}) = O(\|\boldsymbol{x}\|^{-d})$ and $\Delta\varphi(\boldsymbol{x}) = O(\|\boldsymbol{x}\|^{-d-2})$ as $\|\boldsymbol{x}\| \to \infty$. Moreover, $\mathcal{R}\{\varphi\} = \psi$.*

Using the above result we show the functional $\|f\|_{\mathcal{R}}$ given in Definition 1 is well-defined:

**Proposition 9.** *For any $f \in \mathrm{Lip}(\mathbb{R}^d)$, the map $L_f(\psi) := -\gamma_d\langle f,(-\Delta)^{(d+1)/2}\mathcal{R}^*\{\psi\}\rangle$ is finite for all $\psi \in \mathcal{S}(\mathbb{P}^d)$, hence $\|f\|_{\mathcal{R}} = \sup\{L_f(\psi) : \psi \in \mathcal{S}(\mathbb{P}^d), \|\psi\|_\infty \le 1\}$ is a well-defined functional taking on values in $[0,+\infty]$.*

*Proof.* Since $f$ is globally Lipschitz we have $|f(\boldsymbol{x})| = O(\|\boldsymbol{x}\|)$, while for any $\psi \in \mathcal{S}(\mathbb{P}^d)$ we have $|(-\Delta)^{(d+1)/2}\mathcal{R}^*\{\psi\}| = O(\|\boldsymbol{x}\|^{-d-2})$ by Lemma 8, hence $|f(\boldsymbol{x})(-\Delta)^{(d+1)/2}\mathcal{R}^*\{\psi\}(\boldsymbol{x})| = O(\|\boldsymbol{x}\|^{-d-1})$ is absolutely integrable, and so $\langle f,(-\Delta)^{(d+1)/2}\mathcal{R}^*\{\psi\}\rangle$ is finite. If $\langle f,(-\Delta)^{(d+1)/2}\mathcal{R}^*\{\psi\}\rangle \neq 0$, we can choose the sign of $\psi$ so that the inner product is positive, which shows that $\|f\|_{\mathcal{R}} \ge 0$. $\qquad\square$

In Section 4 we showed $\Delta h_\alpha = \mathcal{R}^*\{\alpha\}$ when $\alpha$ was a measure with a smooth density having rapid decay. The next key lemma shows this equality still holds in the sense of distributions when $\alpha$ is any measure in $M(\mathbb{P}^d)$.

**Lemma 9.** *Let $f = h_{\alpha,\boldsymbol{v},c}$ for any $\alpha \in M(\mathbb{P}^d), \boldsymbol{v} \in \mathbb{R}^d, c \in \mathbb{R}$. Then we have $\langle f,\Delta\varphi\rangle = \langle \alpha,\mathcal{R}\{\varphi\}\rangle$ for all $\varphi \in C^\infty(\mathbb{R}^d)$ such that $\varphi(\boldsymbol{x}) = O(\|\boldsymbol{x}\|^{-d})$ and $\Delta\varphi(\boldsymbol{x}) = O(\|\boldsymbol{x}\|^{-d-2})$ as $\|\boldsymbol{x}\| \to \infty$.*

*Proof.* Consider the ridge function $r_{\boldsymbol{w},b}(\boldsymbol{x}) := \frac{1}{2}|\boldsymbol{w}^\top\boldsymbol{x}-b|$, which is generated by the even measure $\alpha_0(\boldsymbol{w}',b') = \frac{1}{2}(\delta(\boldsymbol{w}'-\boldsymbol{w},b'-b)+\delta(\boldsymbol{w}'+\boldsymbol{w},b'+b))$. An easy calculation shows that $\Delta r_{\boldsymbol{w},b}(x) = \delta(\boldsymbol{w}^\top\boldsymbol{x}-b)$ in the sense of distributions, *i.e.*, for all test functions $\varphi \in \mathcal{S}(\mathbb{R}^d)$ we have

$$\int r_{\boldsymbol{w},b}(\boldsymbol{x})\Delta\varphi(\boldsymbol{x})\,d\boldsymbol{x} = \int_{\boldsymbol{w}^\top\boldsymbol{x}=b}\varphi(\boldsymbol{x})\,ds(\boldsymbol{x}) = \mathcal{R}\{\varphi\}(\boldsymbol{w},b). \tag{60}$$

Since $\mathcal{R}\{\varphi\}(\boldsymbol{w},b)$ is well-defined for all $\varphi \in C^\infty(\mathbb{R}^d)$ with decay like $O(\|\boldsymbol{x}\|^{-d})$, by continuity $\Delta r_{\boldsymbol{w},b}(\boldsymbol{x})$ extends uniquely to a distribution acting on this larger space of test functions.

Now consider the more general case of $f = h_\alpha$ with $\alpha \in M(\mathbb{P}^d)$. Then for all $\varphi \in C^\infty(\mathbb{R}^d)$ with $\varphi(\boldsymbol{x}) = O(\|\boldsymbol{x}\|^{-d})$ and $\Delta\varphi(\boldsymbol{x}) = O(\|x\|^{-d-2})$ as $\|\boldsymbol{x}\| \to \infty$ we have

$$\int_{\mathbb{R}^d} f(\boldsymbol{x})\Delta\varphi(\boldsymbol{x})\,d\boldsymbol{x} = \int_{\mathbb{R}^d}\left(\int_{\mathbb{S}^{d-1}\times\mathbb{R}}\frac{1}{2}(|\boldsymbol{w}^\top\boldsymbol{x}-b|-|b|)\,d\alpha(\boldsymbol{w},b)\right)\Delta\varphi(\boldsymbol{x})\,d\boldsymbol{x} \tag{61}$$

$$= \int_{\mathbb{S}^{d-1}\times\mathbb{R}}\left(\int_{\mathbb{R}^d}\frac{1}{2}(|\boldsymbol{w}^\top\boldsymbol{x}-b|-|b|)\Delta\varphi(\boldsymbol{x})\,d\boldsymbol{x}\right)d\alpha(\boldsymbol{w},b) \tag{62}$$

$$= \int_{\mathbb{S}^{d-1}\times\mathbb{R}}\left(\int_{\mathbb{R}^d}r_{\boldsymbol{w},b}(\boldsymbol{x})\Delta\varphi(\boldsymbol{x})\,d\boldsymbol{x}\right)d\alpha(\boldsymbol{w},b) \tag{63}$$

$$= \int_{\mathbb{S}^{d-1}\times\mathbb{R}}\mathcal{R}\{\varphi\}(\boldsymbol{w},b)\,d\alpha(\boldsymbol{w},b) \tag{64}$$

where in (62) we applied Fubini's theorem to exchange the order of integration, whose application is justified since

$$h_{|\alpha|}(\boldsymbol{x}) = \frac{1}{2} \int_{\mathbb{S}^{d-1} \times \mathbb{R}} (|\boldsymbol{w}^\top \boldsymbol{x} - b| - |b|) \, d|\alpha|(\boldsymbol{w}, b) \le \|\alpha\|_1 \|\boldsymbol{x}\| \tag{65}$$

and by assumption $\Delta\varphi(\boldsymbol{x}) = O(\|\boldsymbol{x}\|^{-d-2})$, hence $h_{|\alpha|}(\boldsymbol{x})|\Delta\varphi(\boldsymbol{x})| = O(\|\boldsymbol{x}\|)^{-d-1}$, and so $\int h_{|\alpha|}(\boldsymbol{x})|\Delta\varphi(\boldsymbol{x})| \, d\boldsymbol{x} < \infty$.

Finally, if $f = h_{\alpha,\boldsymbol{v},c}$ for any $\alpha \in M(\mathbb{P}^d)$, $\boldsymbol{v} \in \mathbb{R}^d$, $c \in \mathbb{R}$, since affine functions vanish under the Laplacian we have $\langle f, \Delta\varphi \rangle = \langle h_\alpha, \Delta\varphi \rangle$, reducing this to the previous case, which gives the claim. $\qquad\square$

The following lemma shows $\|f\|_{\mathcal{R}}$ is finite if and only if $f$ is an infinite-width net, in which case $\|f\|_{\mathcal{R}}$ is given by the total variation norm of the unique even measure defining $f$.

**Lemma 10.** *Let $f \in \mathrm{Lip}(\mathbb{R}^d)$. Then $\|f\|_{\mathcal{R}}$ is finite if and only if there exists a unique even measure $\alpha \in M(\mathbb{P}^d)$ and unique $\boldsymbol{v} \in \mathbb{R}^d, c \in \mathbb{R}$ with $f = h_{\alpha,\boldsymbol{v},c}$, in which case $\|f\|_{\mathcal{R}} = \|\alpha\|_1$.*

*Proof.* Suppose $\|f\|_{\mathcal{R}}$ is finite. Then by definition $f$ belongs to $\mathrm{Lip}(\mathbb{R}^d)$ and the linear functional $L_f(\psi) = -\gamma_d\langle f, (-\Delta)^{(d-1)/2}\mathcal{R}^*\{\psi\}\rangle$ is continuous on $\mathcal{S}(\mathbb{P}^d)$ with norm $\|f\|_{\mathcal{R}}$. Since $\mathcal{S}(\mathbb{P}^d)$ is a dense subspace of $C_0(\mathbb{P}^d)$, by continuity there exists a unique extension $\tilde{L}_f$ to all of $C_0(\mathbb{P}^d)$ with the same norm. Hence, by the Riesz representation theorem, there is a unique measure $\alpha \in M(\mathbb{P}^d)$ such that $\tilde{L}_f(\psi) = \int \psi \, d\alpha$ for all $\psi \in C_0(\mathbb{P}^d)$ and $\|f\|_{\mathcal{R}} = \|\alpha\|_1$.

We now show $f = h_{\alpha,\boldsymbol{v},c}$ for some $\boldsymbol{v} \in \mathbb{R}^d, c \in \mathbb{R}$. First, we prove $\Delta f = \Delta h_\alpha$ as tempered distributions (*i.e.*, as linear functionals on the space of Schwartz functions $\mathcal{S}(\mathbb{R}^d)$). By Lemma 9 we have $\langle \Delta h_\alpha, \varphi \rangle = \langle \alpha, \mathcal{R}\{\varphi\}\rangle$ for any $\varphi \in \mathcal{S}(\mathbb{R}^d)$, hence

$$\langle \Delta h_\alpha, \varphi \rangle = \langle \alpha, \mathcal{R}\{\varphi\}\rangle \tag{66}$$
$$= \tilde{L}_f(\mathcal{R}\{\varphi\}) \tag{67}$$
$$= L_f(\mathcal{R}\{\varphi\}) \tag{68}$$
$$= \gamma_d\langle f, (-\Delta)^{(d+1)/2}\mathcal{R}^*\{\mathcal{R}\{\varphi\}\}\rangle \tag{69}$$
$$= -\gamma_d\langle f, \Delta(-\Delta)^{(d-1)/2}\mathcal{R}^*\{\mathcal{R}\{\varphi\}\}\rangle \tag{70}$$
$$= \langle f, \Delta\varphi \rangle \tag{71}$$
$$= \langle \Delta f, \varphi \rangle \tag{72}$$

where in (68) we used the fact that $\mathcal{R}\{\varphi\} \in \mathcal{S}(\mathbb{P}^d)$ for all $\varphi \in \mathcal{S}(\mathbb{R}^d)$ (Helgason, 1999, Theorem 2.4), and in (71) we used the inversion formula for Radon transform: $-\gamma_d(-\Delta)^{(d-1)/2}\mathcal{R}^*\{\mathcal{R}\{\varphi\}\} = \varphi$ for all $\varphi \in \mathcal{S}(\mathbb{R}^d)$ (Helgason, 1999, Theorem 3.1).

Hence, we have shown $\Delta f = \Delta h_\alpha$ as tempered distributions. This means $f - h_\alpha$ is in null space of the Laplacian acting on tempered distributions, which implies $f - h_\alpha = p$ where $p$ is some harmonic polynomial (*i.e.*, $p$ is a polynomial in $\boldsymbol{x} = (x_1, ..., x_d)$ such that $\Delta p(\boldsymbol{x}) = 0$ for all $\boldsymbol{x} \in \mathbb{R}^d$). Finally, since both $f$ and $h_\alpha$ are Lipschitz they have at most linear growth at infinity, so must $p$. This implies $p$ must be an affine function $p(\boldsymbol{x}) = \boldsymbol{v}^\top \boldsymbol{x} + c$, which shows $f = h_{\alpha,\boldsymbol{v},c}$ as claimed.

Conversely, suppose $f = h_{\alpha,\boldsymbol{v},c}$ for some $\alpha \in M(\mathbb{P}^d), \boldsymbol{v} \in \mathbb{R}^d, c \in \mathbb{R}$. Let $\psi \in \mathcal{S}(\mathbb{P}^d)$. By Lemma 8, the function $\varphi = -\gamma_d(-\Delta)^{(d-1)/2}\mathcal{R}^*\{\psi\}$ is in $C^\infty(\mathbb{R}^d)$ with $\varphi(x) = O(\|x\|^{-d})$, $\Delta\varphi(x) = O(\|x\|^{-d-2})$ as $\|x\| \to \infty$, and $\psi = \mathcal{R}\{\varphi\}$. Hence, by Lemma 9 we have

$$L_f(\psi) = \langle f, \Delta\varphi \rangle = \langle \alpha, \mathcal{R}\{\varphi\}\rangle = \langle \alpha, \psi \rangle. \tag{73}$$

This shows

$$\|f\|_{\mathcal{R}} = \sup\{\langle \alpha, \psi \rangle : \psi \in \mathcal{S}(\mathbb{P}^d), \|\psi\|_\infty \le 1\} \tag{74}$$
$$= \sup\{\langle \alpha, \psi \rangle : \psi \in C_0(\mathbb{P}^d), \|\psi\|_\infty \le 1\} \tag{75}$$
$$= \|\alpha\|_1 \tag{76}$$

where the second to last equality holds since $\mathcal{S}(\mathbb{R}^d)$ is a dense subspace of $C_0(\mathbb{R}^d)$, and the last equality is by the dual characterization of the total variation norm.

Finally, to show uniqueness, suppose $h_{\alpha,\boldsymbol{v},c} = h_{\beta,\boldsymbol{v}',c'}$ for some other even $\beta \in M(\mathbb{P}^d)$, $\boldsymbol{v}' \in \mathbb{R}^d$, $c' \in \mathbb{R}$. Then the function $h_{\alpha,\boldsymbol{v},c} - h_{\beta,\boldsymbol{v}',c'} = h_{\alpha-\beta,\boldsymbol{v}-\boldsymbol{v}',c-c'}$ is identically zero, hence by the argument above $\|h_{\alpha-\beta,\boldsymbol{v}-\boldsymbol{v}',c-c'}\|_{\mathcal{R}} = \|\alpha - \beta\|_1 = 0$, which implies $\alpha = \beta$. Therefore, $h_{\alpha,\boldsymbol{v},c} = h_{\alpha,\boldsymbol{v}',c'}$, which also implies $\boldsymbol{v}' = \boldsymbol{v}$ and $c = c'$. $\qquad\square$

Note that Lemma 1 is essentially a corollary of the uniqueness in the preceding result; we give the proof here for completeness.

*Proof of Lemma 1.* Suppose $\overline{R}_1(f)$ is finite. Then by the optimization characterization in Lemma 7, we have $f = h_{\alpha,\boldsymbol{v},c}$ for some even $\alpha \in M(\mathbb{P}^d)$, $\boldsymbol{v} \in \mathbb{R}^d$, $c \in \mathbb{R}^d$, and $\overline{R}_1(f)$ is the minimum of $\|\alpha^+\|_1$ over all even measures $\alpha^+ \in M(\mathbb{P}^d)$ and $\boldsymbol{v}' \in \mathbb{R}^d, c' \in \mathbb{R}$ such that $f = h_{\alpha^+,\boldsymbol{v}',c'}$. By Lemma 10, there is a unique even measure $\alpha^+ \in M(\mathbb{P}^d)$, $\boldsymbol{v} \in \mathbb{R}^d$, and $c \in \mathbb{R}$ such that $f = h_{\alpha^+,\boldsymbol{v},c}$. Hence, $\overline{R}_1(f) = \|\alpha^+\|_1$. $\qquad\square$

Now we give the proof of our main theorem, which shows $\|f\|_{\mathcal{R}} = \overline{R}_1(f)$.

*Proof of Theorem 1.* Suppose $\overline{R}_1(f)$ is finite. By Lemma 1, $\overline{R}_1(f) = \|\alpha\|_1$ where $\alpha \in M(\mathbb{P}^d)$ is the unique even measure such that $f = h_{\alpha,\boldsymbol{v},c}$ for some $\boldsymbol{v} \in \mathbb{R}^d$, $c \in \mathbb{R}$. Furthermore, $\|f\|_{\mathcal{R}} = \|\alpha\|_1$ by Lemma 10. Hence, $\overline{R}_1(f) = \|f\|_{\mathcal{R}}$. Conversely, if $\|f\|_{\mathcal{R}}$ is finite, then by Lemma 10 we have $f = h_{\alpha,\boldsymbol{v},c}$ for a unique even measure $\alpha \in M(\mathbb{P}^d)$, and again by Lemma 1, $\|f\|_{\mathcal{R}} = \|\alpha\|_1 = \overline{R}_1(f)$. $\qquad\square$

*Proof of Proposition 1.* The Radon transform is a bounded linear operator from $L^1(\mathbb{R}^d)$ to $L^1(\mathbb{S}^{d-1} \times \mathbb{R})$ (see, e.g., Boman & Lindskog (2009)). Hence, if $\Delta^{(d+1)/2}f \in L^1(\mathbb{R}^d)$ then $\mathcal{R}\{\Delta^{(d+1)/2}f\} \in L^1(\mathbb{R}^d)$. Let $\alpha \in M(\mathbb{P}^d)$ be the even measure on $\mathbb{S}^{d-1} \times \mathbb{R}$ with density $\gamma_d\mathcal{R}\{\Delta^{(d+1)/2}f\}$. Then $\|\alpha\|_1 = \gamma_d\|\mathcal{R}\{\Delta^{(d+1)/2}f\}\|_1$, *i.e.*, the total variation norm of $\alpha$ coincides with the $L^1$-norm of its density. Therefore, by definition of $\|f\|_{\mathcal{R}}$ we have

$$\|f\|_{\mathcal{R}} = \sup\{\gamma_d\langle f, \Delta^{(d+1)/2}\mathcal{R}^*\{\psi\}\rangle : \psi \in \mathcal{S}(\mathbb{P}^d), \|\psi\|_\infty \le 1\} \tag{77}$$

$$= \sup\{\langle\gamma_d\mathcal{R}\{\Delta^{(d+1)/2}f\}, \psi\rangle : \psi \in \mathcal{S}(\mathbb{P}^d), \|\psi\|_\infty \le 1\} \tag{78}$$

$$= \sup\{\langle\alpha, \psi\rangle : \psi \in \mathcal{S}(\mathbb{P}^d), \|\psi\|_\infty \le 1\} \tag{79}$$

$$= \|\alpha\|_1 = \gamma_d\|\mathcal{R}\{\Delta^{(d+1)/2}f\}\|_1. \tag{80}$$

where we used the fact that the Schwartz class $\mathcal{S}(\mathbb{P}^d)$ is dense in $C_0(\mathbb{P}^d)$ and the dual definition of the total variation norm (37). If additionally $f \in L^1(\mathbb{R}^d)$, we have $\mathcal{R}\{\Delta^{(d+1)/2}f\} = \partial_b^{d+1}\mathcal{R}\{f\}$ by the Fourier slice theorem, which gives $\|f\|_{\mathcal{R}} = \gamma_d\|\partial_b^{d+1}\mathcal{R}\{f\}\|_1$. $\qquad\square$

## E  PROOF OF THEOREM 2

We show how our results change without the addition of the unregularized linear unit $\boldsymbol{v}^\top\boldsymbol{x}$ in (3). Specifically, we want to characterize $\overline{R}(f)$ given in (7) (or equivalently its optimization formulation (9)). Unlike in the univariate setting, $\overline{R}(f)$ does not have a simple closed form expression in higher dimensions. However, for any $f \in \mathrm{Lip}(\mathbb{R}^d)$ we prove the bounds

$$\max\{\|f\|_{\mathcal{R}}, 2\|\nabla f(\infty)\|\} \le \overline{R}(f) \le \|f\|_{\mathcal{R}} + 2\|\nabla f(\infty)\| \tag{81}$$

where the vector $\nabla f(\infty) \in \mathbb{R}^d$ can be thought of as the gradient of the function $f$ "at infinity"; see below for a formal definition. In particular, if $f(\boldsymbol{x})$ vanishes at infinity then $\nabla f(\infty) = \boldsymbol{0}$ and we have $\overline{R}(f) = \|f\|_{\mathcal{R}} = \overline{R}_1(f)$.

For any $f \in \mathrm{Lip}(\mathbb{R}^d)$, define $\nabla f(\infty) \in \mathbb{R}^d$ by[17]

$$\nabla f(\infty) := \lim_{r \to \infty} \frac{1}{c_d r^{d-1}} \oint_{\|\boldsymbol{x}\| = r} \nabla f(\boldsymbol{x}) \, ds(\boldsymbol{x}), \tag{82}$$

where $c_d = \int_{\mathbb{S}^{d-1}} d\boldsymbol{w}$. We will relate $\nabla f(\infty)$ to the "linear part" of an infinite-width net. Towards this end, define $\mathcal{V} : M(\mathbb{S}^{d-1} \times \mathbb{R}) \to \mathbb{R}^d$ to be the linear operator given by

$$\mathcal{V}(\alpha) = \frac{1}{2} \int_{\mathbb{S}^{d-1} \times \mathbb{R}} \boldsymbol{w} \, d\alpha(\boldsymbol{w}, b). \tag{83}$$

Note that if $\alpha = \alpha^+ + \alpha^-$ where $\alpha^+$ is even and $\alpha^-$ is odd, then $\mathcal{V}(\alpha) = \mathcal{V}(\alpha^-)$ since $\int_{\mathbb{S}^{d-1} \times \mathbb{R}} \boldsymbol{w} \, d\alpha^+(\boldsymbol{w}, b) = 0$. In particular, if we set $\boldsymbol{v}_0 = \mathcal{V}(\alpha^-)$, then $h_{\alpha^-}(\boldsymbol{x}) = \boldsymbol{v}_0^\top \boldsymbol{x}$.

**Lemma 11.** *Suppose $f = h_{\alpha,c}$ for any $\alpha \in M(\mathbb{S}^{d-1} \times \mathbb{R})$, $c \in \mathbb{R}$. Then, $\nabla f(\infty) = \mathcal{V}(\alpha)$.*

*Proof.* A simple calculation shows the weak gradient of $f = h_{\alpha,c}$ is given by

$$\nabla f(\boldsymbol{x}) = \int_{\mathbb{S}^{d-1} \times \mathbb{R}} H(\boldsymbol{w}^\top \boldsymbol{x} - b) \boldsymbol{w} \, d\alpha(\boldsymbol{w}, b) \tag{84}$$

where $H$ is defined as $H(t) = 1$ if $t \geq 0$ and $H(t) = 0$ if $t < 0$ otherwise. Therefore, we have

$$\lim_{r \to \infty} \frac{1}{r^{d-1}} \oint_{\|\boldsymbol{x}\| = r} \nabla f(\boldsymbol{x}) \, ds(\boldsymbol{x}) = \lim_{r \to \infty} \int_{\mathbb{S}^{d-1} \times \mathbb{R}} \int_{\mathbb{S}^{d-1}} H(r \boldsymbol{w}^\top \boldsymbol{w}' - b) \boldsymbol{w} \, d\boldsymbol{w}' d\alpha(\boldsymbol{w}, b) \tag{85}$$

$$= \lim_{r \to \infty} \int_{\mathbb{S}^{d-1} \times \mathbb{R}} \boldsymbol{w} \left( \int_{\boldsymbol{w}^\top \boldsymbol{w}' \geq b/r} d\boldsymbol{w}' \right) d\alpha(\boldsymbol{w}, b) \tag{86}$$

$$= \left( \frac{1}{2} \int_{\mathbb{S}^{d-1}} d\boldsymbol{w}' \right) \int_{\mathbb{S}^{d-1} \times \mathbb{R}} \boldsymbol{w} \, d\alpha(\boldsymbol{w}, b) \tag{87}$$

Finally, dividing both sides by $c_d = \int_{\mathbb{S}^{d-1}} d\boldsymbol{w}$ gives the result. $\qquad\square$

**Lemma 12.** *If $f(\boldsymbol{x}) = \boldsymbol{v}_0^\top \boldsymbol{x} + c$ then $\overline{R}(f) = 2\|\boldsymbol{v}_0\|$.*

*Proof.* Note that $f = h_{\alpha,c}$ only if $\alpha$ is odd and $\mathcal{V}(\alpha) = \boldsymbol{v}_0$. Hence, we have

$$\overline{R}(f) = \min_{\alpha \text{ odd}} \|\alpha\|_1 \ \ s.t. \ \ \mathcal{V}(\alpha) = \boldsymbol{v}_0 \tag{88}$$

The adjoint $\mathcal{V}^* : \mathbb{R}^d \to C_b(\mathbb{S}^{d-1} \times \mathbb{R})$ is given by $[\mathcal{V}^* \boldsymbol{y}](\boldsymbol{w}, b) = \frac{1}{2} \boldsymbol{w}^\top \boldsymbol{y}$. Therefore, the dual of the convex program above is given by

$$\max_{\substack{\boldsymbol{y} \in \mathbb{R}^d \\ \|\mathcal{V}^* \boldsymbol{y}\|_\infty \leq 1}} \boldsymbol{v}_0^\top \boldsymbol{y} = \max_{\|\boldsymbol{y}\| \leq 2} \boldsymbol{v}_0^\top \boldsymbol{y} = 2\|\boldsymbol{v}_0\| \tag{89}$$

where we used the fact that $\|\mathcal{V}^* \boldsymbol{y}\|_\infty = \max_{\boldsymbol{w} \in \mathbb{S}^{d-1}} \frac{1}{2} \|\boldsymbol{w}^\top \boldsymbol{y}\| \leq 1$ holds if and only if $\|\boldsymbol{y}\| \leq 2$. This means $2\|\boldsymbol{v}_0\|$ is a lower bound for $\overline{R}(f)$. Since this bound is reached with the primal feasible choice $\alpha$ defined by

$$\alpha(\boldsymbol{w}, b) = \|\boldsymbol{v}_0\| \left( \delta \left( \boldsymbol{w} - \frac{\boldsymbol{v}_0}{\|\boldsymbol{v}_0\|}, b \right) - \delta \left( \boldsymbol{w} + \frac{\boldsymbol{v}_0}{\|\boldsymbol{v}_0\|}, b \right) \right) \tag{90}$$

we have $\overline{R}(f) = 2\|\boldsymbol{v}_0\|$ as claimed. $\qquad\square$

Now we give the proof of Theorem 2.

---

[17]Note every Lipschitz function has a weak gradient $\nabla f \in L^\infty(\mathbb{R}^d)$, so $\nabla f(\infty)$ is well-defined.

*Proof of Theorem 2.* Suppose $\|f\|_{\mathcal{R}}$ is finite. Set $\boldsymbol{v}_0 = \nabla f(\infty)$. Then by Lemma 10, there is a unique even measure $\alpha^+$ such that $f = h_{\alpha^+,\boldsymbol{v}_0,c}$ for some unique $\boldsymbol{v}_0 \in \mathbb{R}^d, c \in \mathbb{R}$, with $\|f\|_{\mathcal{R}} = \|\alpha^+\|_1$. Therefore, $\overline{R}(f)$ is equivalent to the optimization problem

$$\overline{R}(f) = \min_{\alpha^- \text{ odd}} \|\alpha^+ + \alpha^-\|_1 \ \ s.t. \ \ \mathcal{V}(\alpha^-) = \boldsymbol{v}_0 \tag{91}$$

Since $\|\alpha^+ + \alpha^-\|_1 \leq \|\alpha^+\|_1 + \|\alpha^-\|_1$, by Lemma 12 we see that $\overline{R}(f) \leq \|\alpha^+\|_1 + 2\|\boldsymbol{v}_0\|$.

Now we show the lower bound. By Proposition 7 we have $\|\alpha^+ + \alpha^-\|_1 \geq \|\alpha^+\|_1 = \|f\|_{\mathcal{R}}$, which gives $\overline{R}(f) \geq \|f\|_{\mathcal{R}}$. By Proposition 7 we also have $\|\alpha^+ + \alpha^-\|_1 \geq \|\alpha^-\|_1$. By the proof of Lemma 12, the minimum of $\|\alpha^-\|_1$ over all odd $\alpha^-$ such that $\mathcal{V}(\alpha^-) = \boldsymbol{v}_0$ is given by $2\|\boldsymbol{v}_0\| = 2\|\nabla f(\infty)\|$. Therefore, $\overline{R}(f) \geq \max\{\|f\|_{\mathcal{R}}, 2\|\nabla f(\infty)\|\}$, as claimed.

$\square$

Finally, we show there are examples where the upper bound in Theorem 2 is attained.

**Proposition 10.** *There exist $f : \mathbb{R}^d \to \mathbb{R}$ in all dimensions $d$ such that*

$$\overline{R}(f) = \|f\|_{\mathcal{R}} + 2\|\nabla f(\infty)\|. \tag{92}$$

*Proof.* Let $\boldsymbol{w}_+, \boldsymbol{w}_- \in \mathbb{S}^{d-1}$ be orthogonal. Consider $f = h_\alpha$ defined by $\alpha = \alpha^+ + \alpha^-$ with

$$\alpha^+ = \delta(\boldsymbol{w} - \boldsymbol{w}_+, b) + \delta(\boldsymbol{w} + \boldsymbol{w}_+, b) \tag{93}$$
$$\alpha^- = \delta(\boldsymbol{w} - \boldsymbol{w}_-, b) - \delta(\boldsymbol{w} + \boldsymbol{w}_-, b) \tag{94}$$

Hence, $f(\boldsymbol{x}) = |\boldsymbol{w}_+^\top \boldsymbol{x}| + \boldsymbol{w}_-^\top \boldsymbol{x}$ (*e.g.*, in 2-D one such function is $f(x, y) = x + |y|$). Replacing the TV-norm with its dual definition, the dual problem for $\overline{R}(f)$ in this instance is given by:

$$\sup_{\substack{\varphi \in C_0(\mathbb{S}^{d-1} \times \mathbb{R}), \boldsymbol{y} \in \mathbb{R}^d \\ \|\mathcal{V}^* \boldsymbol{y} + \varphi\|_\infty \leq 1}} \boldsymbol{w}_-^\top \boldsymbol{y} + \langle \alpha^+, \varphi \rangle \tag{95}$$

Set $\boldsymbol{y}^* = 2\boldsymbol{w}_-$, and let $\varphi^*$ be a continuous approximation to $\text{sign}(\alpha^+)$ whose support is localized to an arbitrarily small neighborhood of $\pm(\boldsymbol{w}_+, 0)$. Then the pair $(\varphi^*, \boldsymbol{y}^*)$ is dual feasible since

$$\psi(\boldsymbol{w}, b) := [\mathcal{V}^* \boldsymbol{y}^*](\boldsymbol{w}, b) + \varphi^*(\boldsymbol{w}, b) = \boldsymbol{w}^\top \boldsymbol{w}_- + \varphi^*(\boldsymbol{w}, b) = \begin{cases} 1 & \text{if } \boldsymbol{w} = \pm \boldsymbol{w}_+ \text{ and } b = 0 \\ \boldsymbol{w}^\top \boldsymbol{w}_- & \text{else} \end{cases}$$

and so $|\psi(\boldsymbol{w}, b)| \leq 1$. For these choices of $(\varphi^*, \boldsymbol{y}^*)$ the dual objective is $2\|\boldsymbol{w}_-\| + \|f\|_{\mathcal{R}}$, which gives a lower bound on $\overline{R}(f)$. But this is also an upper bound on $\overline{R}(f)$ hence $\overline{R}(f) = \|f\|_{\mathcal{R}} + 2\|\boldsymbol{w}_-\|$. Since $\nabla f(\infty) = \boldsymbol{w}_-$, the result follows. $\square$

## F  Properties of the $\mathcal{R}$-norm

Here we prove the properties of $\mathcal{R}$-norm discuseed in Section 4.1, including Proposition 2.

**Proposition 11.** *The $\mathcal{R}$-norm has the following properties:*

- *(1-homogeneity and triangle inequality) If $\|f\|_{\mathcal{R}}, \|g\|_{\mathcal{R}} < \infty$, then $\|c \cdot f\|_{\mathcal{R}} = |c| \|f\|_{\mathcal{R}}$ for all $c \in \mathbb{R}$ and $\|f + g\|_{\mathcal{R}} \leq \|f\|_{\mathcal{R}} + \|g\|_{\mathcal{R}}$, i.e., $\|\cdot\|_{\mathcal{R}}$ is a semi-norm.*

- *(Annihilation of affine functions) $\|f\|_{\mathcal{R}} = 0$ if and only if $f$ is affine, i.e., $f(\boldsymbol{x}) = \boldsymbol{v}^\top \boldsymbol{x} + c$ for some $\boldsymbol{v} \in \mathbb{R}^d, c \in \mathbb{R}$.*

- *(Translation and rotation invariance) If $g(\boldsymbol{x}) = f(\boldsymbol{U}\boldsymbol{x} + \boldsymbol{y})$ where $\boldsymbol{y} \in \mathbb{R}^d$ and $\boldsymbol{U} \in \mathbb{R}^{d \times d}$ is any orthogonal matrix, then $\|g\|_{\mathcal{R}} = \|f\|_{\mathcal{R}}$.*

- *(Scaling with dilations/contractions) Suppose $\|f\|_{\mathcal{R}} < \infty$. Let $f_\varepsilon(\boldsymbol{x}) := f(\boldsymbol{x}/\varepsilon)$, then $\|f_\varepsilon\|_{\mathcal{R}} = \varepsilon^{-1}\|f\|_{\mathcal{R}}$.*

*Proof.* The 1-homogenity and triangle inequality properties follow immediate from the linearity of all operations and the definition by way of a set supremum.

Clearly $\|f\|_{\mathcal{R}} = 0$ if $f$ is affine. Conversely, suppose $\|f\|_{\mathcal{R}} = 0$ then by the uniqueness in Lemma 10, we have $\alpha = 0$, and so $f = h_{0,\boldsymbol{v},c}$ for some $\boldsymbol{v} \in \mathbb{R}^d$ and $c \in \mathbb{R}$, hence $f$ is affine.

For simplicity we demonstrate proofs of the remaining properties under the same conditions of Proposition 1, *i.e.*, $d$ odd, and where $f$, $\Delta^{(d+1)/2}f \in L^1(\mathbb{R}^d)$ so that $\|f\|_{\mathcal{R}} = \gamma_d\|\mathcal{R}\{\Delta^{(d+1)/2}f\}\|_1 = \gamma_d\|\partial_b^{d+1}\mathcal{R}\{f\}\|_1 < \infty$. The general case follows from standard duality arguments.

To show translation invariance, define $f_{(\boldsymbol{y})}(\boldsymbol{x}) := f(\boldsymbol{x} - \boldsymbol{y})$. Then since $\Delta$ commutes with translations we have $\Delta^{(d+1)/2}f_{(\boldsymbol{y})} = [\Delta^{(d+1)/2}f]_{(\boldsymbol{y})}$. Also, for any function $g$ we see that

$$\mathcal{R}\{g_{(\boldsymbol{y})}\}(\boldsymbol{w}, b) = \mathcal{R}\{g\}(\boldsymbol{w}, b + \boldsymbol{w}^\top \boldsymbol{y}), \tag{96}$$

Therefore,

$$\|f_{(\boldsymbol{y})}\|_{\mathcal{R}} = \int_{\mathbb{S}^{d-1}\times\mathbb{R}} |\mathcal{R}\{\Delta^{(d+1)/2}f_{(\boldsymbol{y})}\}(\boldsymbol{w}, b)|\, d\boldsymbol{w}\, db \tag{97}$$

$$= \int_{\mathbb{S}^{d-1}\times\mathbb{R}} |\mathcal{R}\{\Delta^{(d+1)/2}f\}(\boldsymbol{w}, b + \boldsymbol{w}^\top \boldsymbol{y})|\, d\boldsymbol{w}\, db \tag{98}$$

$$= \int_{\mathbb{S}^{d-1}\times\mathbb{R}} |\mathcal{R}\{\Delta^{(d+1)/2}f\}(\boldsymbol{w}, b)|\, d\boldsymbol{w}\, db = \|f\|_{\mathcal{R}}. \tag{99}$$

To show rotation invariance, let $f_{\boldsymbol{U}}(\boldsymbol{x}) = f(\boldsymbol{U}\boldsymbol{x})$ where $\boldsymbol{U}$ is any orthogonal $d \times d$ matrix. Then, using the fact that the Laplacian commutes with rotations, we have $\Delta^{(d+1)/2}f_{\boldsymbol{U}}(\boldsymbol{x}) = \Delta^{(d+1)/2}f(\boldsymbol{U}\boldsymbol{x})$, and since $\mathcal{R}\{g_{\boldsymbol{U}}\}(\boldsymbol{w}, b) = \mathcal{R}\{g\}(\boldsymbol{U}\boldsymbol{w}, b)$, we see that $\mathcal{R}\{\Delta^{(d+1)/2}f_{\boldsymbol{U}}\}(\boldsymbol{w}, b) = \mathcal{R}\{\Delta^{(d+1)/2}f\}(\boldsymbol{U}\boldsymbol{w}, b)$, and so

$$\|f_{\boldsymbol{U}}\|_{\mathcal{R}} = \|f\|_{\mathcal{R}}. \tag{100}$$

To show the scaling under contractions/dilations (*i.e.*, Proposition 2), let $f_\varepsilon(\boldsymbol{x}) = f(\boldsymbol{x}/\varepsilon)$ for $\varepsilon > 0$. Then

$$\mathcal{R}\{f_\varepsilon\}(\boldsymbol{w}, b) = \int_{\boldsymbol{w}^\top \boldsymbol{x}=b} f(\boldsymbol{x}/\varepsilon)ds(\boldsymbol{x}) \tag{101}$$

$$= \varepsilon^{d-1} \int_{\boldsymbol{w}^\top \tilde{\boldsymbol{x}}=b/\varepsilon} f(\tilde{\boldsymbol{x}})ds(\tilde{\boldsymbol{x}}) \tag{102}$$

$$= \varepsilon^{d-1}\mathcal{R}\{f\}(\boldsymbol{w}, b/\varepsilon). \tag{103}$$

Hence, we have

$$|\partial_b^{d+1}\mathcal{R}\{f_\varepsilon\}(\boldsymbol{w}, b)| = \varepsilon^{d-1}\varepsilon^{-d-1}|\partial_b^{d+1}\mathcal{R}\{f\}(\boldsymbol{w}, b/\varepsilon)| \tag{104}$$

$$= \varepsilon^{-2}|\partial_b^{d+1}\mathcal{R}\{f\}(\boldsymbol{w}, b/\varepsilon)| \tag{105}$$

and so

$$\int_{\mathbb{S}^{d-1}\times\mathbb{R}} |\partial_b^{d+1}\mathcal{R}\{f_\varepsilon\}(\boldsymbol{w}, b)|\, d\boldsymbol{w}\, db = \varepsilon^{-2} \int_{\mathbb{S}^{d-1}\times\mathbb{R}} |\partial_b^{d+1}\mathcal{R}\{f\}(\boldsymbol{w}, b/\varepsilon)|\, d\boldsymbol{w}\, db \tag{106}$$

$$= \varepsilon^{-1} \int_{\mathbb{S}^{d-1}\times\mathbb{R}} |\partial_b^{d+1}\mathcal{R}\{f\}(\boldsymbol{w}, \tilde{b})|\, d\boldsymbol{w}\, d\tilde{b} \tag{107}$$

$$= \varepsilon^{-1}\|f\|_{\mathcal{R}}. \tag{108}$$

$\square$

**Fourier estimates** For any Lipschitz function $f$ we can always interpret $\Delta f$ in a distributional sense. An interesting special case is when $\Delta f$ is a distribution of order zero, *i.e.*, when there exists a constant $C$ such that $|\langle \Delta f, \varphi \rangle| \leq C\|\varphi\|_\infty$ for all smooth compactly supported functions $\varphi$ so that $\Delta f$ extends uniquely to a measure having finite total variation. In this case, the Fourier transform

of $\Delta f$, defined as $\widehat{\Delta f}(\boldsymbol{\xi}) := \langle \Delta f, e^{-j2\pi \boldsymbol{x}^\top \boldsymbol{\xi}} \rangle$ for all $\boldsymbol{\xi} \in \mathbb{R}^d$, is a continuous and bounded function, and we can make use of an extension of the Fourier slice theorem to Radon transforms of measures (see, e.g., Boman & Lindskog (2009)) to analyze properties of $\|f\|_{\mathcal{R}}$. In particular, the following result shows that in order for the $\|f\|_{\mathcal{R}}$ to be finite, the Fourier transform of $\Delta f$ (or the Fourier transform of $f$ if it exists classically) must decay at a dimensionally dependent rate.

**Proposition 12.** *Suppose $\Delta f$ is a distribution of order zero. Then $\|f\|_{\mathcal{R}}$ is finite only if $\widehat{\Delta f}(\sigma \cdot \boldsymbol{w}) = O(|\sigma|^{-(d-1)})$ as $|\sigma| \to \infty$ for all $\boldsymbol{w} \in \mathbb{S}^{d-1}$. If additionally $f \in L^1(\mathbb{R}^d)$, then $\|f\|_{\mathcal{R}}$ is finite only if $\widehat{f}(\sigma \cdot \boldsymbol{w}) = O(|\sigma|^{-(d+1)})$ as $|\sigma| \to \infty$ for all $\boldsymbol{w} \in \mathbb{S}^{d-1}$.*

*Proof.* If $\Delta f \in M(\mathbb{R}^d)$ is a finite measure then its Radon transform $\mathcal{R}\{\Delta f\} \in M(\mathbb{P}^d)$ exists as a finite measure, *i.e.*, we can define $\mathcal{R}\{\Delta f\}$ via duality as $\langle \mathcal{R}\{\Delta f\}, \varphi \rangle = \langle \Delta f, \mathcal{R}^*\{\varphi\} \rangle$ for all $\varphi \in \mathcal{C}_0(\mathbb{R}^d)$ (see, e.g., Boman & Lindskog (2009)). Additionally, the restriction $\mathcal{R}\{\Delta f\}(\boldsymbol{w}, \cdot) \in M(\mathbb{R})$ is a well-defined finite measure for all $\boldsymbol{w} \in \mathbb{S}^{d-1}$, and its 1-D Fourier transform in the $b$ variable is given by

$$\mathcal{F}_b\mathcal{R}\{\Delta f\}(\boldsymbol{w}, \sigma) = \widehat{\Delta f}(\sigma \cdot \boldsymbol{w}) \quad \text{for all } \boldsymbol{w} \in \mathbb{S}^{d-1}, \sigma \in \mathbb{R}. \tag{109}$$

By Lemma 10, $\|f\|_{\mathcal{R}}$ is finite if and only if the functional $L_f(\psi) = -\gamma_d \langle f, (-\Delta)^{(d+1)/2} \mathcal{R}^*\{\psi\} \rangle$ defined for all $\psi \in \mathcal{S}(\mathbb{P}^d)$ extends to a unique measure $\alpha \in M(\mathbb{P}^d)$. We compute the Fourier transform of $\alpha$ in the $b$ variable via duality: for all $\varphi \in \mathcal{S}(\mathbb{P}^d)$ we have

$$\langle \mathcal{F}_b\alpha, \varphi \rangle = \langle \alpha, \mathcal{F}_b\varphi \rangle \tag{110}$$

$$= -\gamma_d \langle f, (-\Delta)^{(d+1)/2} \mathcal{R}^*\{\mathcal{F}_b\varphi\} \rangle \tag{111}$$

$$= \gamma_d \langle \Delta f, (-\Delta)^{(d-1)/2} \mathcal{R}^*\{\mathcal{F}_b\varphi\} \rangle \tag{112}$$

$$= \gamma_d \langle \Delta f, \mathcal{R}^*\{(-\partial_b^2)^{(d-1)/2} \mathcal{F}_b\varphi\} \rangle \tag{113}$$

$$= \gamma_d \langle \Delta f, \mathcal{R}^*\{\mathcal{F}_b(|\sigma|^{d-1}\varphi)\} \rangle \tag{114}$$

$$= \gamma_d \langle \mathcal{R}\{\Delta f\}, \mathcal{F}_b(|\sigma|^{d-1}\varphi) \rangle \tag{115}$$

$$= \gamma_d \langle \mathcal{F}_b\mathcal{R}\{\Delta f\}, |\sigma|^{d-1}\varphi \rangle \tag{116}$$

$$= \gamma_d \langle |\sigma|^{d-1}\mathcal{F}_b\mathcal{R}\{\Delta f\}, \varphi \rangle \tag{117}$$

This shows $\mathcal{F}_b\alpha = \gamma_d|\sigma|^{d-1}\mathcal{F}_b\mathcal{R}\{\Delta f\}$ in the sense of distributions. Since $\mathcal{F}_b\mathcal{R}\{\Delta f\}$ is defined pointwise for all $(\boldsymbol{w}, b) \in \mathbb{S}^{d-1} \times \mathbb{R}$ so is $\mathcal{F}_b\alpha$ and we have

$$\mathcal{F}_b\alpha(\boldsymbol{w}, \sigma) = \gamma_d|\sigma|^{d-1}\mathcal{F}_b\mathcal{R}\{\Delta f\}(\boldsymbol{w}, \sigma) = \gamma_d|\sigma|^{d-1}\widehat{\Delta f}(\sigma \cdot \boldsymbol{w}). \tag{118}$$

Finally, since $\alpha$ is a finite measure, we know $\|\mathcal{F}_b\alpha\|_\infty \leq \|\alpha\|_1 = O(1)$, which gives the first result. If additionally $f \in L^1(\mathbb{R}^d)$ then we have $\widehat{\Delta f}(\boldsymbol{\xi}) = \|\boldsymbol{\xi}\|^2 \widehat{f}(\boldsymbol{\xi})$, and so $(\mathcal{F}_b\alpha)(\boldsymbol{w}, b) = |\sigma|^{d+1}\widehat{f}(\sigma \cdot \boldsymbol{w})$ which gives the second result. $\square$

## G  Upper and Lower bounds

Here we prove several upper and lower bounds for the $\mathcal{R}$-norm. Proposition 3 is an immediate corollary of the following upper bound:

**Proposition 13.** *If $(-\Delta)^{(d+1)/2}f$ is a finite measure, then*

$$\|f\|_{\mathcal{R}} \leq \gamma_d c_d \|(-\Delta)^{(d+1)/2}f\|_1, \tag{119}$$

*In particular, if $(-\Delta)^{(d+1)/2}f$ exists in a weak sense then $\|\cdot\|_1$ can be interpreted as the $L^1$-norm.*

*Proof.* Straight from definitions we have

$$\|f\|_{\mathcal{R}} = \sup\left\{\gamma_d\langle f, (-\Delta)^{(d+1)/2}\mathcal{R}^*\{\psi\}\rangle : \psi \in \mathcal{S}(\mathbb{P}^d), \|\psi\|_\infty \leq 1\right\} \tag{120}$$

$$= \sup\left\{\gamma_d\langle (-\Delta)^{(d+1)/2}f, \mathcal{R}^*\{\psi\}\rangle : \psi \in \mathcal{S}(\mathbb{P}^d), \|\psi\|_\infty \leq 1\right\} \tag{121}$$

$$\leq \sup\left\{\gamma_d\langle (-\Delta)^{(d+1)/2}f, \varphi\rangle : \varphi \in C_0(\mathbb{R}^d), \|\varphi\|_\infty \leq c_d\right\} \tag{122}$$

$$= \gamma_d c_d\|(-\Delta)^{(d+1)/2}f\|_1 \tag{123}$$

where we used the fact that $\mathcal{R}^*\{\varphi\} \in C_0(\mathbb{R}^d)$ for $\varphi \in \mathcal{S}(\mathbb{P}^d)$ (Solmon, 1987, Corollary 3.6) and we have $\|\mathcal{R}^*\{\varphi\}\|_\infty \leq c_d$ for all $\varphi \in \mathcal{S}(\mathbb{P}^d)$ such that $\|\varphi\|_\infty \leq 1$ since

$$|\mathcal{R}^*\{\varphi\}(\boldsymbol{x})| \leq \int_{\mathbb{S}^{d-1}} |\varphi(\boldsymbol{w}, \boldsymbol{w}^\top \boldsymbol{x})| \, d\boldsymbol{w} \leq \int_{\mathbb{S}^{d-1}} d\boldsymbol{w} = c_d. \tag{124}$$

$\square$

The following result also gives a useful lower bound on the $\mathcal{R}$-norm.

**Proposition 14.** *If $f \in \mathrm{Lip}(\mathbb{R}^d)$ then*

$$\|f\|_\mathcal{R} \geq \sup\left\{\langle f, \Delta\varphi\rangle : \varphi \in \mathcal{S}(\mathbb{R}^d), \|\mathcal{R}\{\varphi\}\|_\infty \leq 1\right\}. \tag{125}$$

*Proof.* Let $\mathcal{S}_H(\mathbb{P}^d) \subset \mathcal{S}(\mathbb{P}^d)$ denote the image of $\mathcal{S}(\mathbb{R}^d)$ under the Radon transform. Then

$$\|f\|_\mathcal{R} = \sup\left\{\gamma_d\langle f, (-\Delta)^{(d+1)/2}\mathcal{R}^*\{\psi\}\rangle : \psi \in \mathcal{S}(\mathbb{P}^d), \|\psi\|_\infty \leq 1\right\} \tag{126}$$

$$\geq \sup\left\{\gamma_d\langle f, (-\Delta)^{(d+1)/2}\mathcal{R}^*\{\psi\}\rangle : \psi \in \mathcal{S}_H(\mathbb{P}^d), \|\psi\|_\infty \leq 1\right\} \tag{127}$$

$$= \sup\left\{\gamma_d\langle f, (-\Delta)^{(d+1)/2}\mathcal{R}^*\{\mathcal{R}\{\varphi\}\}\rangle : \varphi \in \mathcal{S}(\mathbb{R}^d), \|\mathcal{R}\{\varphi\}\|_\infty \leq 1\right\} \tag{128}$$

$$= \sup\left\{\langle f, \Delta\varphi\rangle : \varphi \in \mathcal{S}(\mathbb{R}^d), \|\mathcal{R}\{\varphi\}\|_\infty \leq 1\right\} \tag{129}$$

where in the last step we used the inversion formula: $\varphi = \gamma_d(-\Delta)^{(d-1)/2}\mathcal{R}^*\{\mathcal{R}\{\varphi\}\}$ for all $\varphi \in \mathcal{S}(\mathbb{R}^d)$. $\square$

Further simplifying the lower bound above gives the following.

**Proposition 15.** *If $f \in \mathrm{Lip}(\mathbb{R}^d)$ then*

$$\|f\|_\mathcal{R} \geq \sup\left\{\langle f, \Delta\varphi\rangle : \varphi \in \mathcal{S}(\mathbb{R}^d), \|\varphi\|_1 \leq 1\right\}. \tag{130}$$

*In particular, if $\Delta f$ exists in a weak sense then $\|f\|_\mathcal{R} \geq \|\Delta f\|_\infty$.*

*Proof.* If $\|\varphi\|_1 = \int |\varphi(\boldsymbol{x})| \, d\boldsymbol{x} \leq 1$ then clearly $|\mathcal{R}\{\varphi\}(\boldsymbol{w}, b)| = |\int_{\boldsymbol{w}^\top \boldsymbol{x} = b} \varphi(\boldsymbol{x}) ds(\boldsymbol{x})| \leq \int_{\boldsymbol{w}^\top \boldsymbol{x} = b} |\varphi(\boldsymbol{x})| \, ds(\boldsymbol{x}) \leq 1$. Hence $\|\varphi\|_1 \leq 1$ implies $\|\mathcal{R}\{\varphi\}\|_\infty \leq 1$. Combining this with the previous proposition gives the first bound. Additionally, by the dual definition of the $L^\infty$ norm, and since $\mathcal{S}(\mathbb{R}^d)$ is dense in $L^1(\mathbb{R}^d)$, the second bound follows. $\square$

## H    RADIAL BUMP FUNCTIONS

**Proof of Proposition 4.** Assume $f \in L^1(\mathbb{R}^d)$ so that its Radon transform $\mathcal{R}\{f\}$ is well-defined, and for simplicity assume $d$ is odd. Note that for a radially symmetric function we have $\mathcal{R}\{f\}(\boldsymbol{w}, b) = \rho(b)$ for some even function $\rho \in L^1(\mathbb{R})$, *i.e.*, the Radon transform of a radially symmetric function does not depend on the unit direction $\boldsymbol{w} \in \mathbb{S}^{d-1}$. Supposing $\partial^{(d+1)}\rho(b)$ exists either as a function or a measure, we have

$$\|f\|_\mathcal{R} = \gamma_d\|\partial_b^{d+1}\mathcal{R}\{f\}\|_1 = \gamma_d c_d \int |\partial^{d+1}\rho(b)| db, \tag{131}$$

where $c_d = \int_{\mathbb{S}^{d-1}} d\boldsymbol{w} = \frac{2\pi^{d/2}}{\Gamma(d/2)}$.

Now we derive an expression for $\rho(b)$ in terms of $g$. First, since $\rho(b) = \mathcal{R}\{f\}(\boldsymbol{w}, b)$ for any $\boldsymbol{w} \in \mathbb{S}^{d-1}$, we can choose $\boldsymbol{w} = \boldsymbol{e}_1 = (1, 0, ..., 0)$, which gives

$$\rho(b) = \mathcal{R}\{f\}(\boldsymbol{e}_1, b) = \int_{x_1 = b} g(\|\boldsymbol{x}\|) dx_2 \cdots dx_d = \int_{\mathbb{R}^{d-1}} g(\sqrt{b^2 + \|\tilde{\boldsymbol{x}}\|^2}) d\tilde{\boldsymbol{x}} \tag{132}$$

where we have set $\tilde{\boldsymbol{x}} = (x_2, ..., x_d)$. Changing to polar coordinates over $\mathbb{R}^{d-1}$, we have

$$\rho(b) = \int_{\mathbb{R}^{d-1}} g(\sqrt{b^2 + \|\tilde{\boldsymbol{x}}\|^2}) d\tilde{\boldsymbol{x}} = c_{d-1} \int_0^\infty g(\sqrt{b^2 + r^2}) r^{d-2} dr. \tag{133}$$

By the change of variables $t^2 = b^2 + r^2$, $t > 0$, we have

$$\rho(b) = c_{d-1} \int_b^\infty g(t)(t^2 - b^2)^{(d-3)/2} t \, dt. \tag{134}$$

Hence, we see that

$$\|f\|_{\mathcal{R}} = \frac{1}{(d-2)!} \left\| \partial_b^{(d+1)} \left[ \int_b^\infty g(t)(t^2 - b^2)^{(d-3)/2} t \, dt \right] \right\|_1 \tag{135}$$

where we used the fact that $\gamma_d c_d c_{d-1} = \frac{1}{(d-2)!}$.

**Calculations in Example 2.** Let $f(\boldsymbol{x}) = g_{d,k}(\|\boldsymbol{x}\|)$ with $\boldsymbol{x} \in \mathbb{R}^d$ where

$$g_{d,k}(r) = \begin{cases} (1 - r^2)^k & \text{if } 0 \le r < 1 \\ 0 & \text{if } r \ge 1. \end{cases} \tag{136}$$

for any $k > 0$. Then a straightforward calculation using (134) gives

$$\rho(b) = \begin{cases} C_{d,k}(1 - b^2)^{k + \frac{d-1}{2}} & \text{if } |b| < 1 \\ 0 & \text{if } b \ge 1. \end{cases} \tag{137}$$

where $C_{d,k} = \frac{\Gamma((d-3)/2) \cdot \Gamma(1+k)}{2\Gamma((d+1)/2)+k)}$. Hence, we have $\|f\|_{\mathcal{R}}$ finite if and only if $\partial_b^d \rho(b)$ has bounded variation, which is true if and only if $k - d + \frac{d-1}{2} \ge 0$, or equivalently, $k \ge \frac{d+1}{2}$. For example, if $d = 3$ then we need $k \ge 2$ in order for $\|f\|_{\mathcal{R}}$ to be finite, consistent with the previous example.

To illustrate scaling of $\|f\|_{\mathcal{R}}$ with dimension $d$, we set $k = (d+1)/2 + 2 = (d+5)/2$ so that $\rho(b) = C_{d,(d+5)/2}(1 - b^2)^{d+2}$ for $|b| \le 1$ and $\rho(b) = 0$ otherwise. Then we can show that $|\partial^{d+1} \rho(b)| \le |\partial^{d+1} \rho(0)|$ for $|b| \le 1$ and $\partial^{d+1} \rho(b) = 0$ for all $|b| \ge 1$. Therefore,

$$\|f\|_{\mathcal{R}} = \frac{1}{(d-2)!} \int_{-1}^1 |\partial^{d+1} \rho(b)| \le \frac{2}{(d-2)!} |\partial^{d+1} \rho(0)| \tag{138}$$

Performing a binomial expansion of $\rho(b)$ and taking derivatives, we obtain

$$\frac{2}{(d-2)!} |\partial^{d+1} \rho(0)| = 2C_{d,(d+5)/2} \binom{d+2}{(d+1)/2} (d+1)d(d-1) = 2d(d+5) \tag{139}$$

for all odd $d \ge 3$. By the lower bound in Proposition 15, we also have $\|f\|_{\mathcal{R}} \ge \|\Delta f\|_\infty = |\Delta f(\mathbf{0})| = d(d+5)$. Hence $\|f\|_{\mathcal{R}} \sim d^2$.

## I  PIECEWISE LINEAR FUNCTIONS

Here we state and prove a more formal version of Proposition 5. Before stating our result, we will need a few definitions relating to the geometry of piecewise linear functions. Recall that any piecewise linear function (with finitely many pieces) is divided into polyhedral regions separated by a finite number of boundaries. Each boundary is $(d-1)$-dimensional and contained in a unique hyperplane. Hence, with every boundary we can associate the unique (up to sign) unit normal to the hyperplane containing it, which we call the *boundary normal*. Additionally, in the case of compactly supported piecewise linear functions, every boundary set that touches the complement of the support set we call an *outer boundary*, otherwise we call it an *inner boundary*.

**Proposition 16.** *Suppose $f : \mathbb{R}^d \to \mathbb{R}$ is a continuous piecewise linear function with compact support such that one (or both) of the following conditions hold:*

  *(a)  at least one of the boundary normals is not parallel with every other boundary normal, or*

  *(b)  $f$ is everywhere convex (or everywhere concave) when restricted to its support, and at least one of the inner boundary normals is not parallel with all outer boundary normals.*

*Then $f$ has infinite $\mathcal{R}$-norm.*

*Proof.* Assume $f$ is a continuous piecewise linear function with compact support satisfying assumption (a) or (b). Let $B_1, ..., B_n$ denote the boundaries between the regions. Since $f$ is piecewise linear and continuous, the distributional Laplacian $\Delta f$ decomposes into a linear combination of Dirac measures supported on the $d - 1$ dimensional boundary sets $B_k$, *i.e.*, for all smooth test functions $\varphi$ we have

$$\langle \Delta f, \varphi \rangle = \sum_{k=1}^{n} c_k \int_{B_k} \varphi(\boldsymbol{x}) \, ds(\boldsymbol{x}). \tag{140}$$

for some non-zero coefficients $c_k \in \mathbb{R}$, where $ds$ indicates integration with respect to the $d - 1$ dimensional surface measure on $B_k$. In particular, if $B_k$ is the boundary separating neighboring regions $R_p$ and $R_q$, then $c_k = \pm \|\boldsymbol{g}_p - \boldsymbol{g}_q\|$ where $\boldsymbol{g}_p$ and $\boldsymbol{g}_q$ are the gradient vectors of $f$ in the region $R_p$ and $R_q$, respectively, with sign determined by whether the function is locally concave (+) or convex (-) at the boundary. Note that $\Delta f$ is a distribution of order zero, *i.e.*, it can be identified with a measure having finite total variation, and it has a well-defined Fourier transform given by

$$\widehat{\Delta f}(\boldsymbol{\xi}) = \sum_{k=1}^{n} c_k \int_{B_k} e^{-i2\pi \boldsymbol{\xi}^\top \boldsymbol{x}} \, ds(\boldsymbol{x}). \tag{141}$$

We show that $\widehat{\Delta f}(\boldsymbol{\xi})$ violates the necessary decay requirements of Proposition 12 in order for $f$ to have finite $\mathcal{R}$-norm. In particular, we show under both conditions (a) and (b) there exists a $\boldsymbol{w}$ such that $\widehat{\Delta f}(\sigma \cdot \boldsymbol{w})$ is asymptotically constant as $|\sigma| \to \infty$, which gives the claim.

For all $k = 1, ..., n$, let $\boldsymbol{w}_k$ denote a boundary normal to the boundary $B_k$ (*i.e.*, a vector $\boldsymbol{w}_k \in \mathbb{S}^{d-1}$ such that $\boldsymbol{w}_k^\top \boldsymbol{x} = 0$ for all $\boldsymbol{x} \in B_k$, which is unique up to sign).

We first prove the claim under condition (a). Suppose, without loss of generality, that the boundary normal $\boldsymbol{w}_1$ is not parallel with all the others, *i.e.*, $\boldsymbol{w}_1 \neq \boldsymbol{w}_k$ for all $k = 2, ..., n$. We will write

$$\widehat{\Delta f}(\sigma \cdot \boldsymbol{w}_1) = F_1(\sigma) + F_2(\sigma) \tag{142}$$

where $F_1(\sigma) = c_1 \int_{B_1} e^{-i2\pi\sigma \boldsymbol{w}_1^\top \boldsymbol{x}} ds(\boldsymbol{x})$ and $F_2(\sigma) = \sum_{k=2}^{n} c_k \int_{B_k} e^{-i2\pi\sigma \boldsymbol{w}_1^\top \boldsymbol{x}} ds(\boldsymbol{x})$, and give decay estimates for $F_1$ and $F_2$ separately.

First, consider $F_1(\sigma)$. Since $\boldsymbol{w}_1^\top \boldsymbol{x} = 0$ for all $\boldsymbol{x} \in B_1$ we have

$$F_1(\sigma) = \int_{B_1} e^{-i2\pi\sigma \boldsymbol{w}_1^\top \boldsymbol{x}} ds(\boldsymbol{x}) = \int_{B_1} ds(\boldsymbol{x}) = s(B_1), \tag{143}$$

where $s(B_1)$ is the $(d - 1)$-dimensional surface measure of $B_1$. In particular $F(\sigma)$ is a non-zero constant for all $\sigma \in \mathbb{R}$.

Now consider $F_2(\sigma)$. In this case, the integrand of $\int_{B_k} e^{-i2\pi\sigma \boldsymbol{w}_1^\top \boldsymbol{x}} ds(\boldsymbol{x})$ for all $k = 2, ..., n$ is not constant, since by assumption $\boldsymbol{w}_1$ not parallel with any of the boundary normals $\boldsymbol{w}_2, ..., \boldsymbol{w}_n$. By an orthogonal change of coordinates, we can rewrite the surface integral over $B_k$ as a volume integral over a set $\tilde{B}_k$ embedded in $(d - 1)$-dimensional space $\tilde{\boldsymbol{x}} = (\tilde{x}_1, ..., \tilde{x}_{d-1})$, so that $\int_{B_k} e^{-i2\pi\sigma \boldsymbol{w}_j^\top \boldsymbol{x}} ds(\boldsymbol{x}) = \int_{\tilde{B}_k} e^{-i2\pi\sigma \tilde{\boldsymbol{w}}_1^\top \tilde{\boldsymbol{x}}} d\tilde{\boldsymbol{x}}$ for some for some non-zero $\tilde{\boldsymbol{w}}_1 \in \mathbb{R}^{d-1}$. Observe that $g(\tilde{\boldsymbol{x}}) := -\frac{\tilde{\boldsymbol{w}}_1}{i2\pi\sigma\|\tilde{\boldsymbol{w}}_1\|} e^{-i2\pi\sigma \tilde{\boldsymbol{w}}_1^\top \tilde{\boldsymbol{x}}}$ has divergence $\nabla \cdot g(\tilde{\boldsymbol{x}}) = e^{-i2\pi\sigma \tilde{\boldsymbol{w}}_1^\top \tilde{\boldsymbol{x}}}$. Therefore, by the divergence theorem we have

$$\int_{\tilde{B}_k} e^{-i2\pi\sigma \tilde{\boldsymbol{w}}_1^\top \tilde{\boldsymbol{x}}} d\tilde{\boldsymbol{x}} = \int_{\tilde{B}_k} \nabla \cdot g(\tilde{\boldsymbol{x}}) d\tilde{\boldsymbol{x}} \tag{144}$$

$$= \oint_{\partial \tilde{B}_k} g(\tilde{\boldsymbol{x}})^\top \boldsymbol{n}(\tilde{\boldsymbol{x}}) ds(\tilde{\boldsymbol{x}}) \tag{145}$$

$$= -\frac{1}{i2\pi\sigma\|\tilde{\boldsymbol{w}}_1\|} \oint_{\partial \tilde{B}_k} e^{-i2\pi\sigma \tilde{\boldsymbol{w}}_1^\top \tilde{\boldsymbol{x}}} \tilde{\boldsymbol{w}}_1^\top \boldsymbol{n}(\tilde{\boldsymbol{x}}) ds(\tilde{\boldsymbol{x}}) \tag{146}$$

where $\boldsymbol{n}(\tilde{\boldsymbol{x}})$ is the outward unit normal to the boundary $\partial \tilde{B}_k$. This gives the estimate

$$\left| \int_{\tilde{B}_k} e^{-i2\pi\sigma \tilde{\boldsymbol{w}}_1^\top \tilde{\boldsymbol{x}}} d\tilde{\boldsymbol{x}} \right| = O(1/\sigma), \quad |\sigma| \to \infty, \tag{147}$$

which holds for any $k = 2, ..., n$. Therefore, $F_2(\sigma) = \sum_{k=2}^{n} c_k \int_{B_k} e^{-i2\pi\sigma \boldsymbol{w}_i^\top \boldsymbol{x}} ds(\boldsymbol{x}) = O(1/\sigma)$ as $|\sigma| \to \infty$. This shows that $\widehat{\Delta f}(\sigma \cdot \boldsymbol{w}_1) \to c_1 s(B_1)$, *i.e.*, $\widehat{\Delta f}(\sigma \cdot \boldsymbol{w}_1)$ is asymptotically constant, which proves the claim.

Now we prove the claim under condition $(b)$. Without loss of generality, let $\boldsymbol{w}_1$ be an inner boundary normal that is not parallel with any outer boundary normal, and assume $f$ is concave when restricted to its support. Let $I_1$ be the indices of all inner boundary normals parallel with $\boldsymbol{w}_1$ (including itself), let $I_2$ be the indices of all inner boundary normals that are not parallel with $\boldsymbol{w}_1$, and let $O$ be the indices of all outer boundary normals. Then we write

$$\widehat{\Delta f}(\sigma \cdot \boldsymbol{w}_1) = F_{I_1}(\sigma) + F_{I_2}(\sigma) + F_O(\sigma) \tag{148}$$

where $F_{I_1}(\sigma) = \sum_{k \in I_1} c_k \int_{B_k} e^{-i2\pi\sigma \boldsymbol{w}_1^\top \boldsymbol{x}} ds(\boldsymbol{x})$, $F_{I_2}(\sigma) = \sum_{k \in I_2} c_k \int_{B_k} e^{-i2\pi\sigma \boldsymbol{w}_1^\top \boldsymbol{x}} ds(\boldsymbol{x})$, and $F_O(\sigma) = \sum_{k \in O} c_k \int_{B_k} e^{-i2\pi\sigma \boldsymbol{w}_1^\top \boldsymbol{x}} ds(\boldsymbol{x})$. By the same argument as above, we can show $F_{I_1}(\sigma) = \sum_{k \in I_1} c_k s(B_k)$. Since the function is concave when restricted to its support, all of the $c_k$ with $k \in I_1$ are positive, hence the sum $\sum_{k \in I_1} c_k s(B_k)$ is non-zero, which shows $F_{I_1}(\sigma)$ is a non-zero constant for all $\sigma \in \mathbb{R}$. Likewise, by the same argument as above, we can show $F_{I_1}(\sigma) = O(1/\sigma)$ and $F_O(\sigma) = O(1/\sigma)$. Therefore, $\widehat{\Delta f}(\sigma \cdot \boldsymbol{w}_1)$ is asymptotically constant, which proves the claim. $\qquad \square$

