# OpenReview forum: "A Function Space View of Bounded Norm Infinite Width ReLU Nets: The Multivariate Case"
_ICLR.cc/2020/Conference — Accept (Poster)_

### Official Review · AnonReviewer1 · 2019-10-21
**Official Blind Review #1**

**Rating:** 8

**Review:**

The paper studies the function space regularization behavior of learning with an infinite-width ReLU network with a bound on the l2 norm of weights, in arbitrary dimension, extending the univariate study of Savarese et al. (2019).

The authors show that the corresponding regularization function is more or less an L1 norm of the (weak) (d+1)st derivatives of the function, and provide a rigorous formal characterization in terms of the "R-norm", which is expressed via duality through the Radon transform and powers of the Laplacian.

In addition, the paper provides a number of implications of this study, such as approximation results through Sobolev spaces, an analysis of the norm of radial bump functions, and a new type of depth separation result in terms of norm as opposed to width.

Overall, this is a strong paper making several interesting and important contributions for our understanding of the inductive bias of ReLU networks. I thus recommend acceptance.

A few comments:
* is it possible to obtain precise characterizations of interpolating solutions in this setting (other than a mere representer theorem with ReLUs), as done in Savarese et al (2019, Theorem 3.3) for the univariate case?

* perhaps the results of Section 5.1 should be contrasted with those of Bach (2017, e.g. Prop. 5), which only require ~ d/2 derivatives instead of ~ d here, albeit with stronger requirements, for essentially the same functional space (though the approximation result is obtained from an associated RKHS, which is smaller).

* are the results on radial bump functions intended to provide insight on approximation or depth separation? what was the motivation behind this section?

Other minor comments/typos:
- after Prop. 1: "intertwining" appears twice
- eq. (22): missing f in l.h.s.
- eq. (23): is the first minus sign needed?
- before Thm. 1: point to which Appendix
- Section 4.1, "In particular, this is what would happen ... d+1": this should be further explained
- Section 4.1, final paragraph, "in order R-norm to be": rephrase
- Section 5.4, "required norm with three layers is finite": which norm? maybe point to a reference? Also, Example 5 could be explained in further detail
- Section 5.5: what is an RKHS semi-norm? you'd always have ||f|| = 0 => f = 0 in an RKHS, by the reproducing property

**Experience Assessment:**

I have published one or two papers in this area.

**Review Assessment: Checking Correctness Of Derivations And Theory:**

I assessed the sensibility of the derivations and theory.

**Review Assessment: Checking Correctness Of Experiments:**

N/A

**Review Assessment: Thoroughness In Paper Reading:**

I read the paper thoroughly.

---

> ### Author Response · Authors · 2019-11-13
> **Comparison with Bach 2017 and other comments**
>
> Thank you for the positive review and the constructive feedback.
>
> * is it possible to obtain precise characterizations of interpolating solutions in this setting (other than a mere representer theorem with ReLUs), as done in Savarese et al (2019, Theorem 3.3) for the univariate case?
>
> We did pursue this question some, but unfortunately did not come up with a satisfying answer. In the univariate case, it is straightforward to show that a minimum norm solution is given by a linear spline with knots at the sample locations, since the function space norm is essentially the second-order total-variation penalty. In the multivariate case, we know that a minimum norm solution is given by an interpolating piecewise linear function, but due to the more complicated function space description involving a Radon transform, we found it difficult to give a more concise description than this.
>
> * perhaps the results of Section 5.1 should be contrasted with those of Bach (2017, e.g. Prop. 5), which only require ~ d/2 derivatives instead of ~ d here, albeit with stronger requirements, for essentially the same functional space (though the approximation result is obtained from an associated RKHS, which is smaller).
>
> We thank the reviewer for pointing this out. We have added discussion in Section 5.1 comparing our result with Bach 2017, and explaining why ~d order derivatives are necessary in our setting. The difference in derivative order comes from the fact that we consider an L^1-type Sobolev norm (i.e., sum of the L^1 norms of derivatives) and not an L^2-type Sobolev norm, as considered in Bach 2017. For an L^1-type Sobolev norm, the scaling ~d is optimal in the sense that this is the scaling required for a sequence of functions approaching a “point evaluation” (i.e., f(x) = 1 if x=x_0 and 0 otherwise) to have unbounded norm. Whereas, for an L^2-type Sobolev norm, the required scaling for this to occur is ~d/2.
>
> * are the results on radial bump functions intended to provide insight on approximation or depth separation? what was the motivation behind this section?
>
> These results were meant to build intuition for dealing with the R-norm where we can obtain more explicit expressions, and to illustrate how the R-norm scales with dimension for a certain class of bump function, which could be important for future approximation and generalization results. We have added a sentence addressing this at the beginning of Section 5.2.
>
> Other minor comments/typos:
> - after Prop. 1: "intertwining" appears twice
> - eq. (22): missing f in l.h.s.
> - eq. (23): is the first minus sign needed?
> - before Thm. 1: point to which Appendix
> - Section 4.1, "In particular, this is what would happen ... d+1": this should be further explained
> - Section 4.1, final paragraph, "in order R-norm to be": rephrase
> - Section 5.4, "required norm with three layers is finite": which norm? maybe point to a reference? Also, Example 5 could be explained in further detail
> - Section 5.5: what is an RKHS semi-norm? you'd always have ||f|| = 0 => f = 0 in an RKHS, by the reproducing property
>
> Thank you for your careful reading. We have addressed all these issues in the revision.
> In particular, the d+1 scaling of derivatives is described in more detail in Sec. 5.1. And the issue of our claim about RKHS norms versus semi-norms is addressed with a footnote on page 10.
> Finally, you are correct that in Eq. (23) the first minus sign is not needed, but we prefer to leave it there to make subsequent derivations tidier,  such as Example 1, and many proofs in the Appendix.

---

### Official Review · AnonReviewer2 · 2019-10-23
**Official Blind Review #2**

**Rating:** 6

**Review:**

This paper gives characterization of the norm required to approximate a given multivariate function by an infinite-width two-layer neural network. An important result is the relation between Radon-transform and the $\mathcal{R}$-norm. This paper also shows application of the norm on some special case.

I suggest this paper being accepted because it provides new insights into the approximation theory for neural networks. The perspective of norm constraint is different from the traditional approximation theory and may serve as a good contribution to the community.

One question is that: in section 4, the equation (19) is differentiated twice to get the equation (20) containing Dirac delta. Although this is intuitively correct, this seems not a strict derivation to my mathematical background. It would be great if the authors can show the strict definition and derivation presented here.

**Experience Assessment:**

I do not know much about this area.

**Review Assessment: Checking Correctness Of Derivations And Theory:**

I assessed the sensibility of the derivations and theory.

**Review Assessment: Checking Correctness Of Experiments:**

N/A

**Review Assessment: Thoroughness In Paper Reading:**

I read the paper at least twice and used my best judgement in assessing the paper.

---

> ### Author Response · Authors · 2019-11-13
> **strict derivation of equation (19)**
>
> Thank you for your comments. While we use the Dirac delta somewhat informally in equations (19) and (20), this calculation is done rigorously in the proof Lemma 9 in Appendix D. In the revised draft, we now indicate this with a footnote on page 6.

---

### Official Review · AnonReviewer3 · 2019-10-23
**Official Blind Review #3**

**Rating:** 6

**Review:**

In this paper, the author analysis the (approximate) function class generated by an infinite-width network when the Euclidean norm is bounded. They extend the work of Savarese et al. on the univariable function by introducing the Randon Transform and R-norm to this problem.  The authors finally prove that any function in Sobolev space could be (approximately) obtained by a bounded network. The results achieved implies some generalization performance analysis and the induction error. Also, according to the authors, the difference between R-norm and RKHS norm might lead to the distinct from neural networks and kernel methods.

I would recommend accepting this paper since it might give a good insight into understanding the performance of the network beyond the traditional method.

**Experience Assessment:**

I have read many papers in this area.

**Review Assessment: Checking Correctness Of Derivations And Theory:**

I assessed the sensibility of the derivations and theory.

**Review Assessment: Checking Correctness Of Experiments:**

N/A

**Review Assessment: Thoroughness In Paper Reading:**

I made a quick assessment of this paper.

---

### Author Response · Authors · 2019-11-13
**summary of changes in revision**

We thank all the reviewers for their careful reading of the manuscript, and have uploaded a revision based on their feedback. Changes addressing the reviewers comments are indicated by blue text. The main change is an expanded discussion in Section 5.1 regarding the order of smoothness in our Sobolev norm bounds, as requested by Reviewer 1.

Additionally, we have made some small changes to Section 5.3. Previously, we claimed in Proposition 5 that *all* continuous piecewise linear functions with compact support have infinite R-norm. However, there was a flaw in our proof, and we needed to weaken the result slightly. Namely, for the result to hold, we need to make some extra conditions on the boundary sets separating the regions on which the function is linear. We discuss these conditions in detail in Appendix I with an updated proof of the result. These conditions are met for a broad class of piecewise linear functions, including the pyramid function in Example 4, and so the change to Proposition 5 does not affect our depth separation result.

---

### Decision · Program_Chairs · 2019-12-19

**Decision:**

Accept (Poster)

**Comment:**

The article studies the set of functions expressed by a  network with bounded parameters in the limit of large width, relating the required norm to the norm of a transform of the target function, and extending previous work that addressed the univariate case. The article contains a number of observations and consequences. The reviewers were quite positive about this article.